

# Water Resources Management, Technology, and Culture in Ancient Iran

Masoud Saatsaz[1, 2], Aboulfazl Rezaie[1, 2]

[1] Center for Research in Climate Change and Global Warming (CRCC), Institute for Advanced Studies in Basic Sciences (IASBS), P.O. Box 45195-1159, Zanjan, Iran

[2] Department of Earth Sciences, Institute for Advanced Studies in Basic Sciences (IASBS), Zanjan 45137-66731, Iran

**Masoud Saatsaz (Corresponding Author)**

Institute for Advanced Studies in Basic Sciences (IASBS), Zanjan, Iran

Phone: +98 33153777

Mobile: +98 916 615 7603

Email: saatsaz@iasbs.ac.ir

**Aboulfazl Rezaie**

Institute for Advanced Studies in Basic Sciences (IASBS), Zanjan, Iran

Phone: +98 33153779

Mobile: +98 910 804 0780

Email: arezaei@iasbs.ac.ir





## 1   Abstract

Iran is one of the countries facing high water risk because of its geographical features, climate variations, and uneven distribution of water resources. Iranians have practiced different water management strategies at various periods following the region's geo-climatological features, needs, tools, available resources (surface water and groundwater), political stability, economic power, and socio-cultural characteristics. This study is a brief history of water management in Iran from pre-civilization times to the end of the Islamic Golden Age (1219 AD). This study pointed out geo-climatological features have consistently been crucial intrinsic properties controlling water regime, settlement patterns, and other socioeconomic issues. These factors caused the early agricultural communities to emerge in water-rich regions of northwestern, western, and southwestern Iran. By the 4th Millennium BC, while water access became more difficult as population growth, economic activity, and urbanization progress, water resources' systematic development appeared in west and southwest Iran under the Mesopotamian civilization. However, despite all benefits, Mesopotamian water-based technology and administration could not meet all water demands in Iran's arid regions. For these reasons, qanats were developed in Persia by the First Persian Empire (Achaemenid Empire). No doubt, the Achaemenids (550-330 BC) should be regarded as one of the early civilizations that emerged in a land without sufficient rainfall and major rivers. In this time, idle and marginal lands of Iran and neighboring regions of the Middle East, North Africa, and Central Asia could be cultivated through the spread of qanat technology, enabling large groups of peasants to increase crop yields and incomes. After a period of recession during the Seleucid Empire (312-63 BC) and the Parthian Empire (247 BC-224 AD), water resources development gained momentum in the Sassanid era (224-651). In this period, the progress of urbanization was expeditious. Consciously, water resources development in Khuzestan plains (Shushtar and Dezful) was crucial for agricultural intensification, economic expansion, and civilization development. The Sassanids wisely adapted Greek watermills to the complicated topography, limited water availability, and variable climate of Iran to produce food. Although the Iranians practiced a new era of water governance under the Sassanid rule (224-651 AD), chaotic Iran in the Late Sasanian and Early Islamic Period led to severe weaknesses in water-related sectors. After Islam's arrival, the Muslim rulers turned their attention from fighting to set up an Islamic civilization to break the socioeconomic stagnation. To achieve the goal, they opened their scientific doors to science and technology centers. Despite all efforts made during the 8th-12th century, the lack of creativity and investment in promoting water technologies, prioritizing political considerations over social benefits, occurring wars, and poor water management have induced the Iranians to lose their power in developing water resources. In today's Iran, the past water-related problems have aggravated by uneven climate change, population rise, rapid industrialization, urban development, and unprecedented changes in lifestyle. Undoubtedly, solving these problems and moving towards a better future is not possible without addressing the past.

## 1. Introduction

Water is essential not only for life support but for health promotion, food security, economic expansion, social development, political stability, and ecosystem protection. In arid regions like Iran, water resources management for achieving sustainable development is far more significant. To ensure proper water resources development, establishing a rational connection between the available resources (surface water and groundwater), spatiotemporal requirements, geo-climatic conditions, cultural values, legal rights, technological tools, political power, and socioeconomic privileges is a crucial issue. In this context, ancient Iran is one of the first civilizations that actively attend to water resources management, administration, and investment in infrastructure and technology, not merely water availability. Although there has been a long history of water governance in Iran, it now has difficulty in solving current water problems. In this regard, understanding how water-related technology, rules, society, economics, and political systems worked in the past provides us more profound knowledge of how our history shaped the present, allowing us to form our future.

This research investigates various aspects of water resources management, technology, and culture in ancient Iran from prehistoric nations to the end of the Islamic Golden Age (Table 1). This study aims to answer the following questions: (i) How did the ancient Persians develop their water resources?; (ii) Why, when, and where did the early water constructions appear on the Iranian Plateau?; (iii) What role did governments play in managing water?; (iv) How did proper water governance contribute to economic and social progress?; (v) How political tensions affect water sustainability?. The answer to these questions can aid managers in succeeding in water resources development.

Several historical investigations have examined water-related topics of ancient Iran (e.g., Kuros and Labbaf Khaneiki 2007; Mays 2010; Gholikandi et al., 2013; Alemohammad and Gharari 2017; Manual et al.,



2018; Mahmoudian and Mahmoudian 2012; Labbaf Khaneiki 2019; Saatsaz 2019). Most of these researches focus
on (i) water infrastructure such as weirs, dams, qanats, water-mills, (ii) water resources management, (iii)
socioeconomic aspects of water, and (iv) architectural design of water-related systems. One of the main
characteristics of this study is the description of a set of related topics, not only in terms of water management
through government authorization, public participation, scientific collaboration, and infrastructure construction
but also in terms of challenges and opportunities of different Iranian ethnic groups in the alleviation of water
stress. This subject (i.e., covering a range of issues) is not adequately reflected in previous studies. Although all
areas of the matter cannot be mentioned in one research, this article tries to cover some of the basics.

The knowledge about ancient water management and technology in Iran is founded mainly on articles,
books, archaeological observations (texts, tablets, and evidence), and other scientific documents (original or
translated) from different historical periods. However, a small number of ancient clay tablets and administrative
records remain intact, most of which have not been translated. Besides, many archaeological sites have not been
investigated historically or dated with a high degree of confidence. Hence, in some cases, detailed information,
improving our knowledge about ancient water-related systems and management, is still lacking.

**Table 1.** A timeline is showing the significant events in Iran from prehistoric nations to the end of the Islamic Golden
Age

## 2. Study Area Description

The modern state of Iran[1], with an area of 1,648,195 km$^2$ (636,372 sq. mi)[2], is located between 44° 02′ E and 63°
20′ E longitude and 25° 03′ N to 39° 46′ N latitude. The region is bordered [i] to the west by Iraq; [ii] the northwest
by Azerbaijan, Armenia, and Turkey; [iii] the north by the south shore of the Caspian Sea [iv]; the northeast by
Turkmenistan [v]; the east by Afghanistan and Pakistan; and [vi] to the south by the Persian Gulf and Oman Sea
(Figure 1). Approximately 60% of Iran is mountainous, covered by Northern Alborz and Western Zagros
mountain ranges. The remaining parts are enclosed by salt deserts, bare lands, rangelands, forests, cultivable lands,
urban lands, and water bodies. The ground surface elevations have a topography range from -26 in the southwest
plains and the Caspian Sea coastal plains in the north to 5671 m.a.s.l.[3] in High Alborz (Soltani et al., 2016).

**Figure 1.** Map of Iran showing the provinces (After Wikimedia Commons: Iran, administrative divisions under CC-BY-SA
license)

Iran enjoys a broad spectrum of climatological conditions (Perrier and Salkini 2012). Climate variability
is due to the domain extension, geographical situation, topographical variability, and different land position to the
water bodies. The main divisions of climate in Iran are hyper-arid (~35.5% of the total area) in the middle and
southeastern parts, arid (~29.2%) and semi-arid (~20.1%) climates in the middle, southern, and eastern parts,
Mediterranean climate (~5%) in the western regions, and humid to hyper-humid (~10.2%) on the south coast of
the Caspian Sea (Amiri and Eslamian 2010). The average annual temperature is 17.6 °C (Economics 2019). In the
same period, the total annual rainfall is, on average, ~228 mm[4] (one-third of the world's average), of which ~
50% falls in winter (correspond to the minimum water demand), ~23% in spring, ~23% in autumn, and ~4% in
summer (coincided with the maximum water demand) (Saatsaz 2019). From a geographical view, the precipitation
ranges from less than ~20 mm/yr in the southeast, east, and central parts to more than ~1,000 mm/yr in the
southern coasts of the Caspian Sea (Mousavi 2005). Another significant spatiotemporal factor in water availability
is evaporation. The annual average evaporation ranges between 1,500 and 2,000 mm, nearly three times the global
average (IRIMO 2017). A large quantity of the annual rainfall (~70%) rather than being used or percolated. The
rainfall shortage and a high evaporation rate are the primary reason for low water circulation. One of the
consequences is that rivers in Iran are primarily ephemeral with small discharges and have not been distributed
regularly.

## 3. Historical Evolution of Life on the Iranian Plateau

Living on the Iranian Plateau started with the dispersal of early modern humans from Africa, dated between at
least ~90,000 and ~50,000 years ago in the Middle-Paleolithic of the Stone Age (Delson 2019). The oldest-known
artifacts from the Middle-Paleolithic, such as stone tools have been found at the Varvasi Cave (in Kermanshah[5]),
Yafteh Cave (around Khorramabad[6]), Kashafrud (in Khorasan Razavi), and Ganj Par Site (in Rostam Abad,
Gilan), signifying the human existence on the Iranian Plateau (Vigne et al., 2005). During the 8th and the 7th
Millennium BC, the first agricultural communities started to emerge in southwestern, western, and northwestern





Iran, where perennial rivers, rainfall, and fertile alluvial soils allowed agrarian societies to develop (Riehl et al., 2013). Meanwhile, the earliest animal domestication began to occur in the Taurus and Zagros Mountains[7] (Zeder 2008; Helmer et al., 2005). Agriculture and domestication growth caused some community members to engage in off-farm activities such as construction, mining, woodworking, metalworking, trading, stone cutting, and other services. All of these components were essential for early civilizations to emerge (Mountjoy 2005).

**4. Water Resources Management in Prehistoric Iran**

Almost 6,000 years ago, the southwestern and western parts of the Iranian Plateau were a part of the "Fertile Crescent" under the control of the Mesopotamian Civilization (Pollock and Susan 1999). In this region, the low-gradient meandering Karun[8], Karkheh[9], Jarrahi[10], and Dez[11] Rivers flow in vast floodplains, underlain by thick alluvial deposits. Meander cut-offs (oxbow lakes), marshes, and abandoned streams are developed alongside these rivers. This area, however, is susceptible to low, erratic rainfall and drought; only irrigated agriculture was feasible. Hence, to meet full irrigation, the Mesopotamians developed a complex of water systems, including canals with different sizes (one central canal and a network of secondary, tertiary, quaternary, and field canals), head-gate, distributors, regulators, inlets and outlets, weirs, levees, and storage reservoirs (Tamburrino 2010) (Figure 2).

**Figure 2.** Hypothetical layout of a network of irrigation canals in South Mesopotamia

In Mesopotamia, careful management of water was essential for both rural and urban development. Susa[12], one of the oldest-known proto-cities of the Middle East, began to flourish in this region between the Karkheh and Dez Rivers as early as 4395 BC (a calibrated radio-carbon date) (Potts 2016). It seems that the Susa formation came after the abandonment of a nearby city called "Chogha Mish." This city was a well-organized settlement with water wells, wastewater facilities, cesspit[13], and stormwater drainage systems (Alizadeh 2008). Ancient canals and waterways were built around Susa to transfer water from the rivers to the region (Viollet 2017). These rivers carry a considerable sediment load, particularly during rainy periods and great floods[14] (De Graef and Tavernier 2013; Tamburrino 2010). To remove particles from water, in ~1250 BC, the first multi-purpose hydraulic structure was built in the "Chogha Zanbil Complex," the religious center of ancient Elam (Semsar Yazdi and Askar Zadeh 2007; Sadr 2017) (Figure 3). Water is transferred from the Karkhe River to a water storage structure through a ~20 km long open canal (Partani and Heidary 2017). With a capacity of 350 m$^3$, the storage system had 10.70 m length, 7.25 m width, and 4.35 m height (Mahmoudian and Mahmoudian 2012), restricted to an inlet canal, two sidewalls, and a front wall made by baked bricks. Water was stored and desilted in the reservoir and transferred to a basin through nine conduits situated at the bottom of the front wall. Each conduit had two inclined surfaces. The distance between the two neighboring conduits was 0.8 m. As the basin had higher elevation than the conduits, earthy materials were removed from the body of water (Sanizadeh 2008). When the pond was fully recharged, the filtered water was diverted to the temple. In the temple, water was distributed by a network of gutters for worship and purification rituals.

**Figure 3.** The remaining of the water treatment system in "Chogha Zanbil" Complex, Khuzestan, Iran (A Front Side View, B Back Side View)   (Adopted from Naghsh Avaran Toos Consulting Engineers Company, 2013)

In Susa, one of the earliest water-related regulations issued by King Hammurabi[15] were unearthed (Veenhof 1995; Cech 2009). Hammurabi's Code was mainly related to (i) fair distribution of water correspondent to the land size, determining the responsibility of farmers to local communities (e.g., canal construction and maintenance, water delivery, cultivating, water dividing, performing post-harvest duties), and imposing legal penalties on water robbing (diversions of water at upstream), damaging water structures (dams, water canals, barriers), and leaving crops and cattle dry (Kornfeld 2009). As Hammurabi's Code considered the different aspects of water governance, these rules greatly impacted water development and management in proto-cities such as Susa. The Achaemenid archaeological records have not provided any information on water-related state laws and regulations yet. In this period, the source of laws picked from previous traditional laws mostly adopted from the Medians period and Aryan's rules. It seems these regulations were mixed with the customs and laws of other civilizations (Ashur, Greeks, Romans, Babylon, and Egypt) to enrich and improve (Hossieni et al., 2016). In some cases, where laws were not available (e.g., qanats), water was governed by customary laws through trusted community members[16].

Before the Achaemenids' rise, the greatest challenge to integrated water management was the lack of a central government that solves large-scale water-related problems. As life in this era closely relied on irrigated agriculture, major farming communities and proto-cities were growing surrounding surface water resources.



Despite all the benefits, living on floodplains had significant disadvantages in flood events and changes in the
river course. In ancient Iran, floods were mostly in association with the geomorphological and meteorological
characteristics of the region. Although heavy rainfall and melting snow have always been considered the main
flood drivers, the meander pattern of large rivers in Iran could intensify floods. In flat areas like the Khuzestan
plains, the flow velocity of rivers drastically decreases, and river sediments deposit. Sediment deposition causes
the river's rise in elevation, allowing it to flood. Flooding might be caused or exacerbated by the change in land-
use during the Post-Neolithic deforestation and urbanization. There is no doubt, large-scale control of water
required massive hydraulic structures and centralized management. It was the gap that the Iranians could fill
during the Achaemenid civilization.

**5. Water Resources Management in the Achaemenid Civilization**

In 550 BC, Cyrus the Great established the First Persian Empire (the second Iranian empire-based Dynasty after
the Median Dynasty) called the Achaemenid Dynasty (Sampson 2008) in the land of "Pars"[17] or "Persis"[18]. The
Achaemenids built the earliest Persian civilization to be socially organized, politically stable, economically firm,
and militarily strong (Sampson 2008).

In Early Achaemenid Iran, the Persians were semi-nomadic tribes[19][20] guided by unsophisticated tribal
laws and traditions (Shahpour Shahbazi 2011). However, the Persians did not wholly give up the nomadic lifestyle
during the governance of the Achaemenids. In winter, they stayed in Babylon, spent part of spring at Susa, and
went to the cooler Ecbatana[21] in summer (Shahpour Shahbazi 2011). There were two other centers in the heart of
their homeland. One was Pasargadae[22], and the other was Persepolis[23], both located in the Province of Pars[24]
(Yamauchi 1991). Over time, the empire could establish a centralized government, causing a growing number of
people. Population and economic growth increased the need for food and producers, which intensified the need
for water. Water shortage pushed the Achaemenids to build qanat, dam, and canal network areas.

**5.1. Qanat System**
Qanat[25] is a gently sloping tunnel (gallery)[26] with some shaft wells, which transfers groundwater to the earth's
surface (Figure 4). Part of the tunnel excavated through an aquifer's phreatic zone is the water-producing zone,
and another part that just serves to transfer water to the ground surface is the water transport zone (Semsar Yazdi
and Labbaf Khaneiki 2016). At the land surface, water flows continuously, sometimes over long distances, from
the qanat outlet (appearance)[27] to consumption areas through a network of open canals (Anglakis et al., 2016).
The climatological, topographical, hydrogeological, and technological factors control the qanat discharge, varying
from 0.001 to 300 $m^3$ per hour (on average, 60 $m^3$ per hour) (Holden 2019).

**Figure 4.** A simple schematic showing a typical qanat system

A critical step in qanat construction is to find a reliable groundwater resource (Boustani 2008).
Landscape, anomalies in soil color and moisture, seepage pattern, vegetation cover, and spring discharge are
conventional indicators to locate a qanat construction point. Ancients presumably knew groundwater could be
available in foothills, wadies, dry riverbeds, and alluvial fans (Waterhistory 2019). After viewing evidence, the
first and deepest shaft named the "mother-well" is dug by a crew of skilled qanat diggers[28] using only simple hand
tools[29]. The mother-well is sunk to the saturated zone for locating the water table and checking the quality,
quantity, and regularity of the groundwater flow (Kheirabadi 2000; Smith 1953). In this stage, the aesthetic
parameters of water (i.e., temperature, turbidity, color, taste, and odor) are detected by qanat diggers through the
senses. The sense of hearing is occasionally used to detect water movement.

In the next step, along a line between the mother-well and qanat outlet, the crew digs vertical shafts with
a diameter between 80 and 100 cm (Semsar Yazdi and Labbaf Khaneiki 2016). At intervals of ~20 to 200 m, the
shafts are used to remove the excavated materials from the main tunnel, air circulation and provide access for
repair and maintenance. During the digging process, excavated soils are dumped all-over the shaft opening to
prevent entering surface runoff to the shaft and the main tunnel. If the soil is loose and unstable, the tunnel and
shaft lining is necessary to improve the qanat durability.

The mother-well depth depends on the water table depth, qanat length, earth slope, and the owner's
capital for excavation, ranging between 30 m and 100 m (Karki et al., 2017). The qanat length could also be either
short or long, varying from a few hundred meters to ~100 km. In qanat design, the slope is one of the most critical
factors that controls a qanat's length. Gradients from 2/1000 to 5/1000 will typically provide sufficient flow
velocities (Semsar Yazdi and Labbaf Khaneiki 2016). Lands with gentle slopes (nearly horizontal) allow more





lengths and *vice versa*. The qanat length and the mother-well depth are often high in dry regions. For instance, in
Yazd and Kerman, the qanat systems extend between ~30 and 50 km long on the margin of Kavir-e Lut (Lut
Desert) in the south-central part of Iran (English 1968).

The origin and outspread of qanat have been discussed by many researchers such as Kobori (1973),
Wilkinson (1977), Goblot (1979), Lightfood (2000), Boucharlat (2003), Magee (2005), and Semsar Yazdi and
Labbaf Khaneiki (2016). As archaeological records provide little information on the qanat's origin, there is still
no widely accepted hypothesis to explain this topic. The earliest written evidence is some inscriptions on stone[30],
date back 714 BC onwards that describe the qanat's existence in the city of Ulhu (modern Ula[31]) by Sargon II[32].
Goblot (1963) believes the qanat birthplace is the Ulhu city, but its technology spread over the entire Iranian
plateau by the Medes and Achaemenids. This hypothesis has raised some questions. The main issue is why people
in a water-rich region like Ulhu needed to dig qanats (Semsar Yazdi and Labbaf Khaneiki 2016). To answer this
question, Magee's theory (2005) may address the issue. Magee believes a hot and arid climate prevailed in the
Middle East during the Late Second and Early First Millennium BC. The sediment analysis and fossils records in
some parts of Pakistan and Turkey confirm this hypothesis (Lückge et al., 2001). For this reason, qanat seems to
be an adaptive response to climate change and water shortage.

Like other water-related systems, qanat has its own advantages and disadvantages (Table 2). From a
social view, qanats can be considered one of the key indicators to determine how and where ancient people lived.
It seems that most ancient qanats in Iran were constructed in the central, eastern, and southeastern regions of the
Iranian Plateau with a dry climate (e.g., in present-day Kerman, Hormozgan, Sistan and Baluchestan, South
Khorasan, and Yazd provinces) (Figure 5). Although these regions have a lower population density than the
country average, they contain many qanats due to their water shortage. According to Briant (2002), further
expansion of qanat technology in these areas led to the emergence of the whole season agriculture, thus ensuring
an increase in agricultural intensification, food supply, and income. The qanat practice was not usual in water-
rich regions (e.g., Khuzestan, Gilan, and Mazandaran) unless surface water resources were fully exploited or
depleted during long-term droughts.

**Table 2.** Advantages and disadvantages of the qanat system
**Figure 5.** The geographical distribution of qanats in Iran

Unlike a water well, qanat systems did not have a point structure; these systems covered the land of many
hundred farmers with unequal shares (Kobori 1973). Hence, a relationship between the government, local owners,
and sharecroppers were essential. On occasion, the qanats' linear structure caused controversy over the water
distribution from upstream to downstream, especially during water shortage periods (Saatsaz 2019). Besides, the
approximation of the buffer zone along the route of each qanat and assessing the owner's contribution to dig,
maintain, and restore a qanat, was challenging. The water allocation challenges existed not only in the past but
also in the present. The Achaemenids were aware of this and made all efforts to protect, clean up, and rehabilitate
qanats through peaceful collaborations. Because of the people's dependency on qanats, the Achaemenid's
government also performed many remunerations and incentive policies for renovating an abandoned qanat. One
of the motivation policies was the tax exemption for revivalists and their descendants for up to five generations
(Semsar Yazdi and Askarzadeh 2007; Semsar Yazdi and Labbaf Khaneiki 2016). According to Nathanson (2013),
Zoroastrian priests[33] had always encouraged farmers to produce more than their demand through developing their
land and water resources[34]. Undoubtedly, these incentives could encourage people to revive their natural resources
and improve their living environment in collaboration with each other.

## 5. 2. Dam Construction in the Achaemenid Era

The Achaemenid Empire was thinking about qanat development but, for many reasons, tried to construct dams.
As the Achaemenids strengthened, there was an increasing demand for water supply, irrigation, flood control, and
diverting water. Regarding the geopolitical, religious, and climatological reasons, all attention was on Persepolis[35]
and Pasargadae[36] (Figure 6). In Susa and Ecbatana, the situation was different. Ecbatana[37] enjoyed abundant
rainfall, allowing the non-irrigated cultivation. In Susa, water-related infrastructures had already been built by
previous civilizations. Undoubtedly, many other water-related facilities had been constructed by the Achaemenids
across Iran, which have either not been studied thoroughly, or destroyed utterly.

**Figure 6.** Geographical location of Persepolis and Pasargadae (A) and the Achaemenid dams (B) in the Marvdasht Plain, Fars
Province (Figure B is based on the global SRTM DEM created by Schacht et al., 2012)





Pasargadae, where Cyrus the Great performed his coronation, lies on the Marvdasht Plain[38] in present-
294  day Fars Province. In a straight line, Persepolis is 40 km to the southwest of Pasargadae with an altitude of 1,770
295  m.a.s.l. (Godard 1962). At the beginning of the spring season, when the Persians celebrated their New Year[39] in
Persepolis, the Marvdasht Plain enjoyed a mild and pleasant climate. However, unlike the pleasant spring, the
plain's climate, based on today's weather conditions, is semi-arid, with an average annual rainfall of 343 mm
(Table 3). The Kor River runs permanently from the northwest to southeast across the Marvdasht Plain and
discharges into the Bakhtegan Lake. The Pulvar Stream[40], the main tributary of Kor River, flows through the plain
from northeast to south-southwest and flows into Kor River at Khan Bridge[41] (Shahpour Shahbazi 2011).

**Table 3.** The average monthly and annual rainfall in the Marvdasht Plain (1973-2016)

Since the Marvdasht Plain's river level was lower than the surrounding areas, it was impossible to use
any water from the streams without artificial assistance and technical installations. Also, the drainage system in
mountainous regions was poor, and most basins experienced abrupt floods. The steep slopes of hillsides and small-
scale alluvial fans, where the channel conveyance capacity of rivers is low, increase the risk of flash floods. Canal
destruction and sedimentation were among critical problems of floods so that the abandonment of water canals
and building new ones were more comfortable than fixing them. Massive networks of diversion and irrigation
canals were needed to divert floodwater and irrigate croplands. Thus, the Achaemenids established many dams,
reservoirs, and networks of water canals to keep rivers safe, store floodwater, divert water, and supply water.

In Fars Province, the "Ramjerd" Dam, "Dariush Dam"[42], "Bande-Sang Dokhtaran", and a collection of
five other dams, including the "Sad-e Alafi-1 Dam," "Sad-e Alafi-2 Dam," "Sad-e Tang-e Saadatashahr Dam,"
"Sad-e Shahidabad Dam," and "Sad-e Didegan Dam," were constructed by the Achaemenids on the Kor River
headstream (Ertsen and Schacht 2013; Schacht et al. 2012, Karami and Talebiyan 2015; Mays 2010). Except for
the Sad-e Shahidabad Dam, situated on a perennial river, the rests are now in dry riverbeds (Schacht et al. 2012).
The Sad-e Didegan Dam is an embankment dam with a watershed area of ~46 km$^2$, constructed in the early
Achaemenid period by earth materials. The dam dimensions are ~105 m in width, 21 m in height, and a crown
length of ~105 m (Schacht et al. 2012)[43]. There are traces of a recharging waterway in the upper parts and a
control structure used to stabilize water flow (Ertsen and Schacht 2013). In the dam's architecture, regular stone
blocks were used, connected by swallow-tailed iron clamps, coated by molten lead [44] (Shahpour Shahbazi 2011).
All the stones were local and quarried on the spot. The dam structure is very similar to "Sad-el Kafarathe Dam,"
built in ~the 3$^{rd}$ Millennium BC by the ancient Egyptians for flood control (Smith 1971).

Another major dam, Sad-e Shahidabad, was built on the Polvar River in the "Tangeh Bolaghi Area" in
the Fars Province (Ertsen and Schacht 2013). This dam has dimensions of ~590 m in width, 15 m high, and a
crown length of ~700 m, where its watershed has an area of ~4,900 km$^2$. Both the Sad-e Didegan and Sad-e
Shahidabad dams have similar canal design and control structures (Schacht et al., 2012). Considering the
traditional form of stones, precise engineering, and unique architectural system, it is clear that the engineers who
constructed Achaemenid dams had enough experience to consider various engineering parameters for dam
construction.

The Achaemenids realized that rainfall and rivers in their territory are insufficient for a secure water
supply. Like other ancient civilizations, the Achaemenids used water-lifting devices for irrigation and domestic
water supply. Hand-operated or animal-powered water-lifting machines were common in Iran. The water lifting
rate for a typical animal-powered waterwheel varied between 480 m$^3$/d (1.5 m height of water lift) and 240 m$^3$/d
(9 m height of water lift) (Molenaar 1956). In some cases, however, well and qanat construction was not
economically or technically feasible. Therefore, they sometimes supplied water from spring sources through
subterranean or open canals. For instance, there was a long underground canal in the Persepolis Complex to
transfer springs' water (Mays 2010). Waterways were usually excavated into the hill's slope, which dominated the
platform to collect and convey rainwater from the mountain to the straightforward, avoiding damage to the
complex (Shahpour Shahbazi 2011). In individual sections, canals were coated with tar to prevent water seepage
beneath the Persepolis platform. Besides, the Achaemenids build levees for flood protection, water-mills for
grinding wheat, canals for water transport, and reservoirs for storing water (Mays 2010).

The building of the Marvdasht historical complex and its surrounding hydraulic structures show how the
Achaemenids could establish a strong link between science, technology, and culture. They had an excellent
background and high understanding of hydrology, civil engineering, and other related sciences such as climatic
hazards, mining and, urban planning. From the climatological point of view, they knew rainstorm season in Fars
starts at the beginning of November and finishes at the end of April. The snow-melting period begins in March
and ends in May. The heavy floods probably occurred in March and April when the snow-melting process follows



rain storms. In the period between March and April, two of the oldest festivals in Iran, known as "Nowruz" and Farvardinegan[45] (The Remembrance Day), were held. As the flood events could have disrupted the ceremonies, building dams and relevant water systems were necessary.

Undoubtedly, the Achaemenids had many other reasons for building the dam. As found in an Old Persian cuneiform text, Darius the Great asks God to protect the land of Persia from the lie, enemy, and famine. The drought meant water shortages, and water shortages meant starvation for them. Assuming the past climatic conditions equal to that in the present, we tried to find the cause of these concerns. First, the moving average of precipitation for three, five, and seven years have been calculated in a period between 1973 and 2016 (e.g., three years moving average of rainfall for a specific year is equal to the average total rainfall in that year, the previous year and the following year) (see Appendix). In the next step, maximum duration, magnitude, intensity, and severity for drought events were determined (Table 4).

**Table 4.** Maximum duration, magnitude, intensity, and severity of drought in the Marvdasht Plain

Based on the PNPI[46] classification (Willeke et al., 1994), the severity of drought can be very slight (80 <P≤ 100), slight (70<P≤ 80), moderate (55 <P≤ 70), severe (40<P≤ 55), and very severe (P≤ 40), of the average rainfall in statistical years. Regarding drought duration, this method also classifies droughts into very slight (0-1 year), slight (2 years), moderate (3 years), severe (4-5) years, and very severe (≥6 years). Table 4 shows that the average maximum severity and duration of droughts in the Marvdasht Plain are classified as moderate and very severe. By occurring such drought, however, the climate of the area moves towards an unfavorable arid regime. If such conditions prevail over the whole country, the climate in more than 65% of the country changes towards a hyper-arid regime. Under such circumstances, drought will be a serious threat to food security. This justifies the Achaemenids' great attention to dams.

The Achaemenids, regardless of all the weaknesses they had at the end of their 220-year reign, could make great achievements, particularly in water-related sectors. Some of their achievements were destroyed by natural and man-made disasters, some were reconstructed by other empires and labeled by their names, and some are still in use. However, the most important achievement of the Achaemenids is creating "national identity"; a concept that water played a crucial role in shaping it.

## 6. Water Resources Management in the Seleucids Era and Parthian Era

Following the conquest of Iran by Alexander the Great in 330 BC, the Iranian satraps[47] were governed by various Greek Satraps forming the Hellenistic Seleucid Empire and then Parthian Empire[48] (Curtis 2007). After the conquest of Iran by Alexander, qanats seem to have been abandoned or destroyed (Ashrafi and Safdarian 2015). Moreover, since the Parthian government was remarkably decentralized, the Parthians were not concerned about the loss of qanats and other hydraulic structures. According to Semsar Yazdi (2006), some qanat systems and irrigational systems were abandoned or damaged. Polybius, a Greek historian of the Hellenistic period[49], recorded that Arsaces III, one of the Parthian kings, tried to destroy some qanats and interrupt water flow to make it difficult for the Seleucids to advance towards the Parthian capital[50] (Beaumont 1971).

## 7. Water Resources Management in the Sassanid Era

The Sassanid's regulations had excellent attention to groundwater, especially to issues concerning the management of qanat. The Sasanian Empire realized that water resources' administration provides them the strength, stability, and durability. Hence, they established the first specific department of water called "Diwan-e Kastfezoud" (also named "Diwan-e Kast-Afzoud")[51] (Ali Abadi 2005). Developing, managing, and protecting water resources, and collecting water tax and tribute from all the territories, constituting rules, solving water-related conflicts were among the department's duties. In this respect, a set of 150 legal documents, written down in the Pahlavi language, related to judgments, contracts and possessions, tax receipts, and lists of the farmland properties has been discovered, translated, and printed, confirming the ability of the Sassanids in structuring their domain (Rezakhani 2008).

### 7. 1. Weir-Bridge Construction in the Sassanid Era

The Sasanians tried to be an urban dynasty through the building and re-building of many cities. They constructed many weir-bridges[52] in both Persian and Roman styles (Table 5). The doctrine of urbanization allowed them to acclimatize with Roman technology. Meanwhile, trade played a significant economic and socio-cultural





role in the development of the cities. At this time, Shushtar[53] and Dezful[54], because of their geographical situation,
mighty rivers, and agricultural lands, had a unique chance for development[55].
The first multipurpose weir-bridge, called Band-e Kaisar"[56], was built by the Sassanid's in the north-west
part of Shushtar over the Shoteit River, the main branch of the Karun River[57]. This weir was used as a bridge for
passing, regulating water level, and diverting water to the Dariyon River during water level rises in the Karun
(Encyclopædia Iranica 1986). It had 43 little arches, 44 central arches, 543 m long, 10-15 m wide, and 8 m high,
built by a combination of sandstone blocks, river stones (Pebbles), mortar, and metal clamps. The basic structure
and material used in this bridge show the bridge was designed and constructed with Roman soldiers' labors,
captured after Valerian's defeat at the Battle of Edessa in 260 AD (Saeidian 2013). Band-e Mizan is one other
well-kept Sassanid weir that diverts the Karun River water to its branches (i.e., Gargar[58] and Shoteit[59]) with a
proportion of two to four, respectively. The weir includes nine sluices (mouths) of different sizes[60], made of cut
sandstones with mortar branches. Some records show both the Mizan and Kaisar weirs were renovated by the
Safavids[61] (1501-1736) and Qajars[62] (1873-1909).
**Table 5.** List of dams constructed by the Sassanids
**7.2. Water-Mills  Building in the Sassanid Era**
Water-mills[63] are among the earliest hydro-technological structures used by the Iranians to facilitate grinding
grains. Earlier grinding was mostly accomplished by animal power[64]; windmills were not typical[65]. Before the
advent of water-mills, peasants were forced to wait for a long time to grind their grains[66]. In the presence of water-
mills as machine-driven, cost-reducing, income-generating, time-saving, and high capacity technology, villagers
could increase the size of their lands, and millers were capable of mass grinding.
The early spread of water-mills in Iran dates back to the Sassanids, especially at the time of King Shapur
I[67], Shapur II[68], Kavad I[69], and Khosrow I[70] (Djamaly et al., 2017; Neely 2011). In this era, farming and agriculture
were the basements of the economy. In this context, water-mills were one of the most significant components of
an intricate network between local water suppliers, grain producers, processors, and consumers. These fulfilled
many roles in economic expansion, urbanization, and rural development. The Sasanians' knowledge and
experiences in hydraulic structure design made it possible for them to generate power using water-flows.
Since the Sassanid Empire, "Greek Mill" and "Roman Mill" have been used to meet the needs. The so-
called "Roman Mill" features a vertical wheel, rotating about a horizontal shaft. Unlike the Roman type, a "Greek
Mill" is powered by a horizontal wheel, rotating around a vertical axle or shaft, without setting up gears. This type
is generally powered by small water volumes directed at high-velocity (Weaver and Pinder 1963). An inclined
aqueduct diverts a proportion of water from a river toward the water-mill in these mills. From a height of one to
20 m, the water drops into a reverse cone-shaped water-tower to provide a pressure head for driving the wheel.
At the bottom of the water-tower a convergent nozzle with varying cross-sectional areas is used to eject the water
to the mill wheel. The flow volume and velocity depend on the water-tower[71] and nozzle diameters[72]. The force
of rushing water keeps the wheel and runner stone turning around. The bedstone is fixed and more resistant than
impact forces than the runner stone[73]. There is a central hole[74] in the turning stone by which the grains fall into
the gap between millstones. The fineness or coarseness of the grind is determined by the gap size and turning
speed. The turning stone speed is dependent on many factors, such as the size of millstones[75], wheel design, and
water discharge (Figure 7).
**Figure 7.** Structure of a typical horizontal water-mill in Iran
Greek mills have been so welcomed by the Iranians (Saeidian 2012). Greek mills are simple, low cost,
and easy to construct, operate, maintain, and repair. Besides, these mills are more secure than Roman ones against
seasonal fluctuations in river discharge and flash flood damages. However, such mills have the disadvantage of
low efficiency, only ~15 to 40%[76]. Hence, these machines were used to mill a small number of grains (Pourjafar
et al., 2010).
Archaeologically, the most robust evidence for the Sassanid's investment in building mills is available in
the "Shushtar Historical Hydraulic System[77]" (Figure 8). Located on the east side of Shushtar; there is a cluster of
~40 water-mills along the Garger River[78] (Harverson 1993). These structures consist of one or two domed rooms
and narrow corridors made of cut sandstone and baked brick (UNESCO 2008). The mills are fed by three tunnels
called Boleyti, Dahan-e Shahr, and Se-Kureh. Although parts of the mills were lost over time, the remains were
renovated recently as "Shushtar Cultural Heritage" to attract tourists.





Greek water-mills, such as those that were constructed in Khuzestan, Ilam, Fars, and Khorasan, were built below weirs. A typical style was pair water-mills in which two sets of water-mills, with one headrace, were used in two neighboring rooms separated by a wall. This mill was designed for grinding two kinds of grains at the same time. In fast-flowing permanent rivers, a string of water-tower mills, fed by a small canals system, was occasionally constructed at irregular intervals ranging between ∼50 m to 1,500 m (Neely 2011; Harverson 1993). The remnants of a string of 22 pre-Islamic water-towers, covering a total distance of ∼6.5 km, are traceable in the Dehlorān Plain[79] (Neely 2011). Other examples can be seen in Jiroft[80] (50 mills), Nishabur[81] (40 mills), and Hamadan (20 mills) (Harverson 2004).

**Figure 8.** Historical hydraulic structures of the Karun River in Shushtar districts, Khuzestan Province (Adopted from UNESCO MAP of Shushtar under CC-BY-SA license)

In some arid regions of Iran, where large permanent rivers are lacking, one or several water-tower mills receive waterpower from a qanat system. Such hybrid systems are built in qanat with sufficient slope and flow velocity near the lower end of the tunnel. The sudden drop of water from the water-tower provides a large driving force for water to transport. As qanat water-mills need the elevation difference to turn the water wheel, the water-mill should be constructed under the qanat's tunnel to enable full water force. Some of these mills are visible in Dehloran (Neely 2011), Ardestan[82] (Harverson 1993), Kashan[83] (UNFAO 2014), Meybod[84] (Saeidian 2013), Taft[85] (Papoli Yazdi and Labbaf Khaneiki 2004), Aradakan[86] (Papoli Yazdi and Labbaf Khaneiki 2000), Kerman[87], and Sarvestan[88] (Harverson 1993).

Qanat-based water-mills can be regarded as an appropriate technology for sustainable development. They have strengthened agricultural livelihoods and food security in central and eastern Iran, where water-milling capacity is inadequate to meet needs. This technology has given local farmers more significant control over their time, cost, and final pricing of their production. In addition to grinding, the qanat mills had other functions such as: (i) increasing water velocity to move towards agricultural lands, (ii) decreasing water temperature and evaporation rate, and (iii) covering the qanat's operation and maintenance costs[89].

In Iran, Roman water-mills have been mostly constructed along large rivers, such as "Zayandeh-Rud[90]", and Karun. Occasionally, a complex of Roman water-mills was built in different sections of a river corridor. Midstream water-mills were operated in dry seasons and riverside in both wet and dry seasons. Roman water-mills was customarily set into two primary levels; a basement for housing the drive system (wheel-house) and a top floor for millstones (grinding room). The grinding room roof was occasionally domed, allowing the air to circulate and light to transmit through the dome openings. The packs of grains were stored in an attic, connected to a hopper to pour grains into the millstones. One of the oldest stream mills[91], dating back to the Sassanid Empire, was constructed in Dezful City[92], at the downstream side of the "Sassanid Bridge[93]"along the Dez River (Eghtedari, 1974; Saeidian 2012).

Through watermills, the Sassanids could introduce a cost-effective, eco-friendly, and sustainable technology to the Iranians. Flour made by a watermill was tasty and fresh; it kept for years without spoiling. It was very common for a mill to be used for centuries. If one mill was severely damaged, another mill would be built on the site. Until the middle of the 20th century, watermills played a crucial role in the country's socioeconomic development. Before World War II, Iran was a special exporter of grain, but in 1941 it faced a severe famine. More deprived people wanted to solve their economic problems, such as eliminating inflation and food supply, especially flour and bread. Maybe from this point, the idea of extensive reforms crossed the Second Pahlavi's mind. After the "White Revolution ", he made a rapid change in economy, lifestyle, and urbanization. Traditional watermills failed to guarantee an adequate supply of flour and disappeared due to technological advancement. In Iran, a small number of watermills are still producing flour. Two of the well-known ones are the Kakhak Watermill in Khorasan Razavi Province and Askzar Watermill in the Yazd Province. The number of operating watermills in Iran is very small compared to Afghanistan, India, and Nepal. Given that old watermills are still seen in many cities (Table 6), these systems can be used to generate green energy after rebuilding and reviving.

**Table 6.** A list showing the location of existing watermill heritage sites in Iran

**8. Water Resources Management in the Golden Age of Islam**
Although the Sasanian's Era was a golden age for the Iranians in terms of agricultural activity, urban development, and economic expansion, it was followed by a tough transitional period, particularly in southwestern and western



Iran[94], the central part of "the Sassanid Empire's agricultural backbone" (Maresca 2019). The exhaustion of the
Iranian army through Sassanid-Byzantine wars (602–628), destroying industry, infrastructure, and civilian
property hand in hand with unprecedented levels of public criticism over economic and social imbalances, were
the main reasons behind the Sassanid Empire fall and the subsequent Islamic conquest of Iran (Rezakhani 2017).
The sharp decline in agricultural production led to a reduction in the country's tax revenue. Decreased attention
to the country's water infrastructure caused severe floods. In total, the food and economic security of the country
was severely endangered. The Sassanids declined like a living creature that decays at the end of its life.

Immediately after the arrival of Islam, Iran had a messy and disorganized environment. Muslims tried to
change the religious, political, institutional, and social structure of the country. The implementation of Islamic
customs[95] and laws[96-97] was one of the first steps towards the Islamization of the society. In the meantime, water
could be an essential link between custom, religion, law, and community, but there were obstacles problems in
the Muslims' path. In the sources of sharia, there were only some concepts such as justice, fairness, and balance,
for the benefit of all societies (Naff 2009). Although the Quran[98] has 63 references to water (Farshad and Zinck
1998), it does not assert any clear duty or rule on water supply and consumption (Absar 2013). The lack or
insufficiency of fundamental rights and obligations regarding access to water, sanitation, sharing, and selling
water was the main barrier to the Islamization of water-related rules. As a case, Arab Muslims had no law or
regulation about qanats because qanat was native to Iran and spread from Iran to neighboring countries.

In cases, there were some contradictions between Islamic rules and traditional customs. In the Islamic
view, water, land, and crops as indivisible, interrelated, and interdependent properties. According to the precepts
of sharia, water cannot be possessed by anyone; it is a free substance, and beyond private ownership, no price
should be paid to use it, and it cannot be sold. Riparian water rights for allocating water have commonly been
limited to amounts measured adequate for a particular crop area (Naff 2009). Such a condition was in stark contrast
to the Sassanid's system. The Sassanid Empire had a rigid social stratification in which social classes differed in
terms of dignity, rank, right, ownership, and control of sources, wealth, and social activities (Aarab 2016). In this
system, nobles and priests lived in a luxurious form, incomparable to a farmer's life. This form was utterly different
from that of Islam that emphasized justice, equality, and fairness. To establish an Islamic system, great flexibility
was needed to reach a compromise with Iranians. In some cases, Muslim jurists had to ignore their laws or make
slight changes in former Iranian laws (Wilkinson 1990).

Although agriculture remained the base of economy and society in the early Islamic period, investment
in agricultural and water infrastructure declined. The differentiation between Muslims and non-Muslims[99],
destruction and abandonment of water infrastructure during wars[100], the disintegration of the administrative
structures, and changes in rules and regulations were the main reasons for the weakening of agriculture in the age
of transition (Soroush 2014; Daniel 2020). However, by strengthening Islam's foundations in Iran, the Muslim
rulers focused on the agricultural sector development as the basement for economic stabilization.

In a long period between the 8[th] to the end of the 12[th] century[101], the Muslim world underwent a golden
age of advancement in science, agriculture, economy, art, architecture, and literature (Saliba 1995). During the
period, Muslims increased their scientific collaboration with Greek, Roman, Chinese, and Hindu scholars[102]. At
that time, water-related sciences were one of the most attractive fields for Iranian scientists. Numerous
documentary and archeological records show the efforts of elites in the Samanid Empire (819-999 AD), Buyid
Dynasty (934-1062), Ghaznavid Empire (962-1186 AD), and Seljuk Empire (1016-1153 AD) to solve water-
related problems (Petersen 1996; Savory 2007; Bastanirad 2012). One of the first texts on hydrology is a book
entitled "*The Extraction of Hidden Waters*" written by the Iranian mathematician and engineer "Muhammad Al-
Karaji" (935-1029 AD), as late as ~1,000 years ago[103] (Nadji and Voight 1972; Al-Hassan and Hill 1986;
Abattouy 1999). In this book, the author addressed different types and origins of waters, exploring groundwater
in drylands, approximating the groundwater depth, digging wells, constructing qanats, estimating the protection
area around qanats, water-related laws, field investigations, and instrumental innovations. In 1014 AD,
Avicenna[104], the brilliant Iranian scientist, in his book titled "The Canon of Medicine"[105], provided some
explanations about the quality of water and the distribution of diseases by water and soil (Mohamed 2008). Nearly
at the same time, another Iranian scientist named "Abu Raihan Muhammad al-Biruni" (973-1048 AD), in his
books entitled "*The Remaining Signs of Past Centuries*"[106]; "Alberuni's India[107]"; "*A Critical Study of What India
Says, Whether Accepted by Reason or Refused*"; and the Mas'udi Law, provided some fundamental explanations
on various bodies of water and the artesian water (Yousif 2000).

During the period, new water infrastructures were built, and old ones were reconstructed. Among the
small dams and bands that were built in this period, the Buyids dams of "Qur'an Gate,"[108] "Band-e Air," "the
Ghaznavid's dams of "Feiz Abad" and "Tous"[109], and the Ilkhanate's dam of "Kebar"[110] can be mentioned





(Tanchev 2014; Norouz and Noorzad 2015). During this area, qanat's technology expanded toward more than 34
countries under different names (English 1968; Behnia 2000; Habashiani 2011) (Table 7). Despite all efforts
made during this period, the lack of creativity and investment in promoting water-related infrastructure and
technologies, occurring wars and territorial conflicts, prioritizing economic and political concerns over social
benefits along with poor water governance have resulted in water insecurity over centuries.

**Table 7.** The historical spread of the qanat under different names (in parentheses)

**9. Conclusion**
In Iran, geo-climatological features are crucial intrinsic properties controlling water regimes, settlement patterns,
and other socioeconomic issues. These factors caused the early agricultural communities to emerge in the fertile
and water-rich regions of southwestern, western, and northern Iran. This trend is currently observed in Iran. The
population distribution, social progress, and economic development in the present-day country are unbalanced
and influenced by many factors such as the climatological features (e.g., rate, duration, and distribution of rainfall),
soil fertility, and availability of surface water (e.g., perennial rivers). In this regard, the water resources
development in the eastern regions has always been quite different from the west. From the social point of view,
today's Iran is most similar to the Sassanid era. In both periods, Iran has experienced rapid growth in population,
urbanization, and food demand. At these points, agricultural activities have been crucial for national development.
The governments emphasized water development in southwestern and western parts; they paid little attention to
the eastern regions. This policy has not without social and environmental consequences. The conversion of
wetlands, pastures, meadows, and other permanent grasslands to irrigated lands in the west and socioeconomic
inequality in the east of Iran have always been among the consequences of uneven water resource development.

War and territorial conflicts have been common challenges between the past and the present. Wars have
led to diminished attention to the country's entire sectors, causing a mess and destabilization. Complete or partial
destruction of water infrastructures and services has caused environmental, economic, and social collapses on a
local and nationwide scale. The direct and indirect consequences of war are: (i) soil-water degradation; (ii)
increasing water allocation conflicts; (iii) endangering public health; (iv) agricultural losses; (v) decreasing rural
family income; (vi) decrease in tax revenues from agricultural products, and (vii) forced migration. Over the last
century, Iran was embroiled in two prolonged civil wars: Iran's Anglo-Soviet invasion (1941) and the Iran-Iraq
War (1980-1988). The eight-year Iran-Iraq War was similar in location to the Arab-Sassanid War. During these
wars, most water-related systems in southwestern and western Iran were ruined, damaged, or polluted, creating a
chaotic environment for developing water resources.

Another problem that continued to the Pahlavi Era (1925-79) was Iran's nomadic culture, making
centralized water resources management difficult. To some extent, the Pahlavi Kingdom, by land and water
allocation, enabled nomadic tribes to have a settled life and engage in social activities. Although they provided
basic facilities to nomads for permanent settlement, there was little attention to teach nomads appropriately the
settled life culture. For this reason, when they touched the least tension in life, they abandoned their farms and
became slum-dwellers.

This study shows that, except for the Achaemenids and Sassanids, leaders and policy-makers could not
stimulate the Iranians to innovate and enhance their water technologies, services, or management practices. During
the Islamic Golden Age, the Iranians' focus was mainly on science's theoretical development; they did not solve
water-related problems practically with the times. For instance, the Iranians did not try to reduce the systematic
disadvantages of the qanat over time. Following an acceleration in population growth, industry expansion, lifestyle
change, and urbanization, the qanat system was ineffective. It was unavoidably marginalized and swapped by
pumping wells. As leaders and policy-makers became familiar with modern agricultural technologies, they paid
attention only to the positive side of modernization without considering the negative side. The unresolved
problems related to irrigation efficiency, crop yield, crop-water requirement, poor water distribution, and water
pollution were piled up over the years, without any motivation to solve them.

There are many other lessons to learn. In ancient Iran, water-related problems were solved by basic
concepts of Hydraulics. In the same way, water-related infrastructures were built using locally available materials.
Still, these managing practices and technology constituted the necessary foundations for today's water governance.
However, although water rights, fairwater allocation, pricing plan, sustainable use, public service, social
responsibility, quality criteria, social benefits, use efficiency, water integrity, and water governance have been
highly regarded in modern sciences, combining these concepts with the traditional ones makes them more





efficient. Accordingly, restoration, stabilization, and upgrading of ancient infrastructures and techniques are necessary before they become forgotten. To sum up, although the future prediction is challenging, the future will be more predictable if the past is adequately recognized.

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





**Table 1.** A timeline is showing the significant events in Iran from prehistoric nations to the end of the Islamic Golden Age.

| Era | Time | Event |
|---|---|---|
| **Pre-Islamic Era** | ~70,000 BC | The first practice of living on the Iranian Plateau. |
| | ~10,000 - 7,000 BC | The first agricultural communities started to emerge in western and northwestern Iran. |
| | ~4,400 BC | The first great proto-cities on the Iranian Plateau grew up in alluvial plains. |
| | 1,250 BC | The oldest water supplying system in Iran was founded in "Chogha Zanbil," an ancient Elamite Complex in southwestern Iran. |
| | ~700 BC | Many ancient Iranian tribes, who settled in northwestern Iran, joined together to make the Median Monarchy. |
| | 550 BC | Cyrus the Great took over the Median Empire and formed one of the most well-known ancient civilizations called the Achaemenid Empire. |
| | 334 - 331 BC | Alexander the Great captured Iran, destroying thousands of qanats and irrigation systems. |
| | 325 BC–224 BC | The Greeks ruled over Iran through the Seleucid Empire and Parthians. In this period, some qanat and irrigational systems were abundant or damaged. |
| | 224 BC - 642 AD | The Sassanids established the first department of water named "Dīwān-e Kastfezoud." In this era, many dams and weirs were constructed and rebuilt or their mills repaired. |
| | 642 AD | The implementation of Islamic customs and laws in water-related affairs. |
| **Islamic Era** | 8th AD - 13th AD | Iranians experienced a "Golden Age" of science and showed keen interest in assimilating other nations' scientific knowledge, writing and translating books. |



**Table 2.** Advantages and disadvantages of the qanat system

|  | **Advantage** | **Disadvantage** |
|---|---|---|
| **Social and Government Aspects** | Qanat enabled nomadic tribes to have a settled life and engage in social activities. | Creating controversy over the approximation of the buffer zone, water allocation and distribution |
|  | Application to transport water over long distances; Allowing the government to utilize barren lands purposefully. |  |
|  | Making a significant relationship between the government, local owners, and farmers for constructing, maintaining and reviving qanats. |  |
|  | Applying digging-related experiences in the military to build underground tunnels for smuggling and defensive purposes; allowing the Achaemenids to extend their authority to farther regions |  |
| **Agricultural and Economic Aspects** | The emergence of the whole season agriculture | To be relatively time-consuming, labor-intensive, and expensive for the construction, maintenance, and repair of qanats |
|  | The increase in agricultural products, food supply, and income; allowing the people to be empowered socially and economically. |  |
|  | Proving service to many caravans, on oases along the Silk Route, developing economic, and cultural trade. |  |
| **Hydrological Aspects** | Insensitive to seasonal and other short-time changes in weather | Having a non-stop discharge during all seasons |
|  | Extracting groundwater as a renewable resource without making rapid drawdown in the aquifer | Not possible to construct in flat areas |
|  | Supplying cold freshwater with low turbidity, and water loss. | Extreme floods and earthquakes can severely damage, or obliterate the qanat shafts and tunnel. |
|  | Using the energy of gravity for water transferring without the need to pump or other forms of energy. |  |
|  | Providing energy through watermills |  |
|  | Collecting surface runoff through the vertical shafts and reduce the risk of flash floods. |  |
|  | The ability to store the qanat water into small reservoirs for later use. |  |



**Table 3.** The average monthly and annual rainfall in the Marvdasht Plain (1973-2016)

| Month | Jan | Feb | Mar | Apr | May | Jun | Jul | Aug | Sep | Oct | Nov | Dec | |
|---|---|---|---|---|---|---|---|---|---|---|---|---|---|
| **Avg. Rainfall (mm)** | 58.8 | 53 | 54.9 | 25.7 | 0.6 | 0.6 | 0.1 | 0 | 0.6 | 17.2 | 61.5 | 70.9 | **Total (mm/Year)** <br> 243.9 |
| **Avg. Temperature (°C)** | 12.5 | 11.7 | 19.5 | 22.1 | 28.7 | 34.4 | 37.9 | 36.9 | 32.2 | 24.1 | 20.6 | 15.7 | **Avg. Annual Temperature (°C)** <br> 24.7 |



**Table 4.** Maximum duration, magnitude, intensity and severity of drought in the Marvdasht Plain

|  | Average annual rainfall | 3 years moving average of rainfall | 5 years moving average of rainfall | 7 years moving average of rainfall | Average of the total |
|---|---|---|---|---|---|
| **Max. Duration (in years)** | 5 | 5 | 6 | 6 | 5.5 |
| **Max. Magnitude (in mm)** | -281.6 | -182 | -182.9 | -146 | -198.1 |
| **Max. Intensity (in mm/year)** | -212 | -104 | -36.4 | -24.3 | -94.1 |
| **Max. Severity** | -212 | -126 | -58.2 | -30.4 | -106.65 |
| **Avg. Recurrence Interval of Droughts (in years)** | 2.3 | 4 | 4.4 | 8.3 | 4.75 |





**Table 5.** List of dams constructed by the Sassanids

| Name | River | Description |
|------|-------|-------------|
| Polband-e Dokhtar | Karun River (Shushtar) | Polband-e Dokhtar is one of the largest Sassanid Weir-bridge in the western part of Iran, built over the Kashkan River. The weir, with a length of about 720 m and height of about 30 m, is made of brick, dressed stone, rubble, lime mortar (in piers and foundations), gypsum mortar (in arches), mud, metal clamps (iron and lead), and Sarooj[1]. The weir-bridge was a part of the Royal road, extended from Istakhr[2] and Ctesiphon[3]. |
| Band-e Kaisar (Valerian Weir- Shadorwan Weir- Polband-e Shadorvan) | Shuteyt River (Shushtar) | Band-e Kaisar, on the north-west side of Shushtar, was built on the rocky bed of Shuteyt River, from east to west, nearly 300 m east of Mizan Weir. |
| Band-e Mizan | Gargar River (Shushtar) | Mizan Weir, with a length of 390 m and a height of 4.5 m, was built in the form of diagonal walls, in the North of Shushtar to the diverts the Karun River water to its branches (i.e., Gargar and Shoteit). Remained walls confirm the existence of watermills in the past on the weir's eastern section. In the western part of the weir, an octagonal tower named "Kolah Farangi Tower" is built to monitor the process of weir design and construction. An octagonal tower called "Kolah Farangi Tower" is created to monitor the operation of weir design and construction. |
| Band-e Gargar | Gargar River (Shushtar) | Gargar Weir, at the northeast of Shushtar, is extending from east to west. The weir dimensions are 83 m long, 12 m wide, and 6 m high. This weir is constructed to divert the Gargar River water to the watermills, residential areas, and irrigation canals. The Gargar connects the Karun in Band-e Ghir, 44 km south of Shushtar. Gargar Weir was renovated in the Safavid Era. |
| Band-e Borj-e-'Ayār (Sabei Kosh) | Gargar River (Shushtar) | Borj-e-'Ayār weir, 7.30 m long and 3.50 m wide, lies across the Gargar River, at the southeast of Shushtar. There is a pond related to the Sabein (Mandaeists[4]) Temple, several historic watermills, and related canals around the weir. The weir was constructed to raise the water level in irrigation canals and provide water for watermills and temple. At present, a small part of the weir is preserved, and other parts are ruined due to road construction. |
| Band-e Khoda Afarin (Band-e Mahibazan) | Dariyon River (Shushtar) | Khoda Afarin, with a length of 500 m, and width-height of two m, was built south of Shushtar to bring up the water level in irrigation canals and link between two sides of the Dariyon River. |
| Band-e Lashgar (Polband-e Darvâzeh) | Dariyon River (Shushtar) | One of the famous hydraulic structures attributed to the Sassanids is Lashkar Weir, which is set up to divert water to the lands in the south of Shushtar. This structure has 104 m in length, eight m in width, and 11 gates, stand on solid columns of mortar, brick, and stone. |
| Band-e Sharabdar | Dariyon River (Shushtar) | Sharabdar Weir, with a length of 35 m, a width of 2 m, and a height of one m, lies in an east-west direction across Raghat Stream[5]. This weir has been built to adjust the water level in irrigation networks. |
| Band-e Kavar (Band-e Kuar, Band-e Bahman) | Qara Aghaj River (Kavar) | This weir is located along the Shiraz-Firuzabad Road in Fars Province, spanning the Qara Aqaj River that flows towards the Persian Gulf. With a length of about 130 m and a height of about 9 m, the weir was constructed to raise the Qara Aghaj water level and direct its flow to Kavar Plain through a canal built in the weir's eastern corner. The weir materials are pieces of natural mountain stone and mortar. |
| Polband-e Dezful | Dez River (Dezful) | Dezful weir-bridge, with 22 arches, was set up over the Dez River to link the western and eastern parts of the city and provide water for agricultural areas and gardens of Dezful. Although the weir strong and durable structure, it was substantially damaged by a great flood in 1903. |
| Band-e Khak | Dariyon River (Shushtar) | Khak Weir, at the southwest of Shushtar, was constructed to prevent the Dariyon River and its neighboring plains from flooding and divert water to its branches. This weir was damaged during road construction activities. |
| Band-e Ahvaz | Karun River (Ahvaz) | This weir is located across the Ahvaz Anticline over the Karun River. The weir collapsed at an unknown time in antiquity. At present, only the wall bases of the weir and traces of mills on the end walls of the weir have remained. |

---

[1] A traditional water-resistant mortar made of clay and lime mixed in a six-to-four ratio (in some cases also mixed with sand, Typha fibers, goat hair, straw, and ashes in specific proportions) (Camões et al., 2012).
[2] "Istakhr" or "Estakhr" was the capital of the Sasanian Dynasty, located five km north of Persepolis.
[3] Ctesiphon was a royal capital of the Parthian and Sassanids, located along the Tigris, 32 km southeast of Baghdad.
[4] Mandaeists follow a monotheistic and gnostic religion, living around rivers in the southeast of Iraq and southwest of Iran.
[5] The Raghat Stream is one of the branches of the Dariyon River.



**Table 6.** A list showing some of existing watermill heritage sites in Iran

| Province | City |
|---|---|
| East Azerbaijan | Jolfa |
| Bushehr | Dashtestan, Dashti, Deyr, Asaluyeh, Kangan |
| Chaharmahal and Bakhtiari | Shahr-e Kord, Koohrang |
| Fars | Sarvestan, Jahrom, Eqlid, Estahban, Darab, Nayriz, Bavanat, Larestan, Qir and Karzin, Khorrambid, Lamerd, Kazerun, Fasa, Firuzabad, Zarrin Dasht, Mamasani, Shiraz, Marvdasht, Sepidan, Pasargad, Mohr. |
| Gilan | Siahkal |
| Hamadan | Malayer |
| Hormozghan | Hajji Abad, Bastak |
| Ilam | Ilam, Chardavol, Darreh Shahr, Deh Luran |
| Isfahan | Aran va Bidgol, Ardestan, Isfahan, Meymeh, Khansar, Kashan, Mobarakeh, Nain, Najafabad, Tiran and Karvan, Natanz |
| Kerman | Kerman, Rigan, Kouhbanan, Zarand |
| Kermanshah | Dalahu, Kermanshah, Kangavar, Gilan-e Gharb, |
| South Khorasan | Ferdows, Birjand, Boshruyeh, Tabas, Nehbandan, Zirkuh |
| Khorasan Razavi | Mashad, Taybad, Khaf, Kashmar, Gonabad, Nishapur, Bajestan, Sabzevar |
| North Khorasan | Bojnurd, Jajarm, Maneh and Samalqan |
| Kuzestan | Deful, Shushtar, Andimeshk, Behbahan |
| Kohgiluyeh and Boyer-Ahmad | Dena, Boyer-Ahmad (Yasuj), Gachsaran |
| Lorestan | Khorramabad, Aligudarz, Dorud, Kuhdasht, Azna |
| | |
| Kordestan | Bijar, Saqqez, Diwandarreh, Qorveh |
| Markazi | Zarandieh, Saveh, Mahalat, Arak, Khomein |
| Mazandaran | Behshahr, Dodangeh |
| Qazvin | Buin Zahra, Qazvin |
| Zanjan | Khodabandeh (Deh Shir) |
| Tehran | Robat Karim, Tehran, Shahr-e-Rey |
| Yazd | Yazd, Mehriz, Meybod, Ashkezar, Ardakan, Bafq, Meybod |
| Sistan and Baluchestan | Zabol, Zahedan, Khash |



**Table 7.** The historical spread of qanat under different names (in parentheses)

| Continent | Country |
|---|---|
| Asia | Iraq (Qanat), Bahrain, Oman, United Arab Emirates, Saudi Arabia, Palestine, Jordan, Oman (Falaj for single and Aflaj for plural), Syria (Qanat Romani), Yemen (Felledj, Ghail, Miyan), Afghanistan (Kariz), Pakistan (Kariz or Kahn in Balochi), China (Karez, Kanjing), Japan (Mambo, Mappo), Korea (Ma-nan-po), Kazakhstan, Azerbaijan (Su lağımı), India (Karez, Nahars, Kundi-Bhandara), Mongolia, and Armenia (Kahreze) |
| Africa | Libya, Algeria (Foggara), Egypt, Tunis, Morocco (Khattara, and Rhettara) |
| Europe | Cyprus, Greece, England, France, Germany, Nederlands, Spain (Galerias, Paquio, Galerías, minas or viajes de agua), Canary Islands (Galerias, Paquio), Italy (Ingruttato for single and Ingruttati for plural), Croatia (Kanata), and Russia |
| South America | Chile, Mexico, Peru, and Barizila (Galerias, Paquio) |



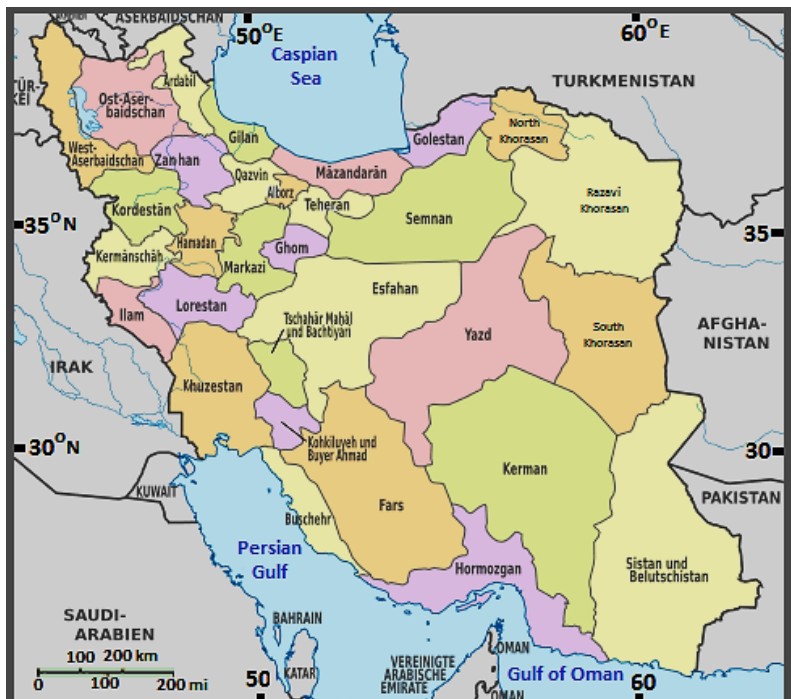

**Figure 1.** Map of Iran showing the provinces (From Wikimedia Commons: Iran, administrative divisions under CC-BY-SA license)





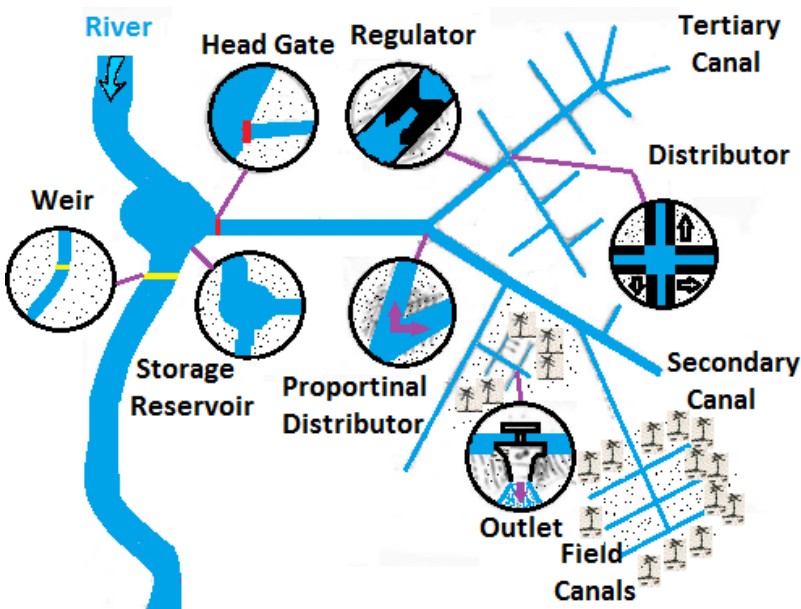

**Figure 2.** Hypothetical layout of a network of irrigation canals in South Mesopotamia





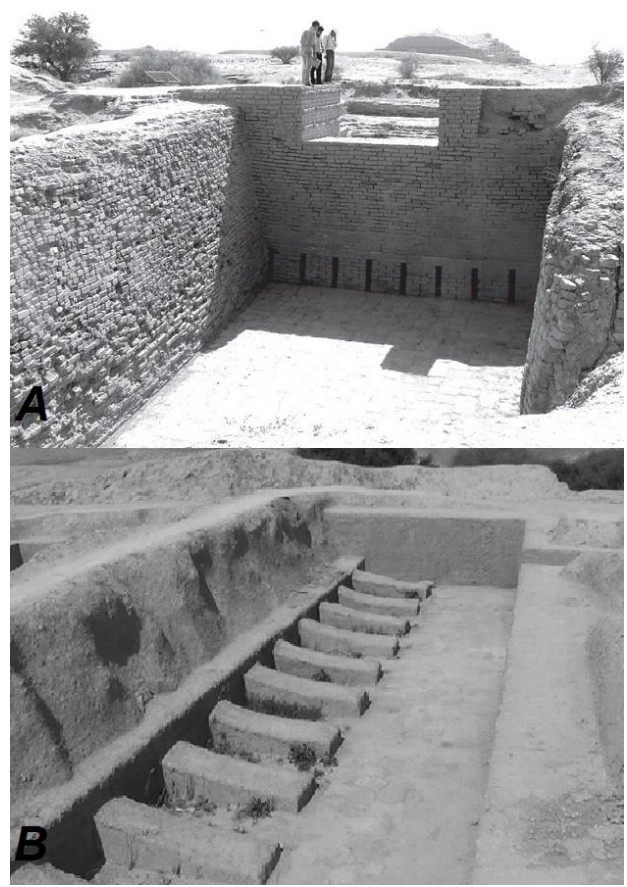

**Figure 3.** The remaining of the water treatment system in "Chogha Zanbil" Complex, Khuzestan, Iran; (A) Front Side View, (B) Back Side View (Adopted from Naghsh Avaran Toos Consulting Engineers Company, 2013).

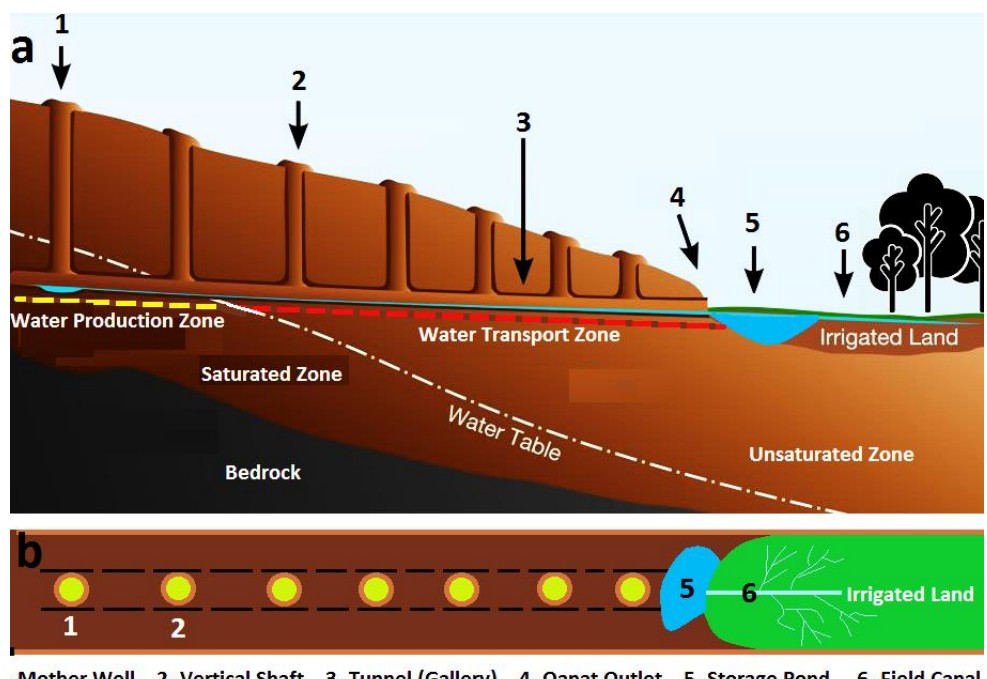

**Figure 4.** A simple schematic showing a typical qanat system; (a) Cross section, (b) Aerial view





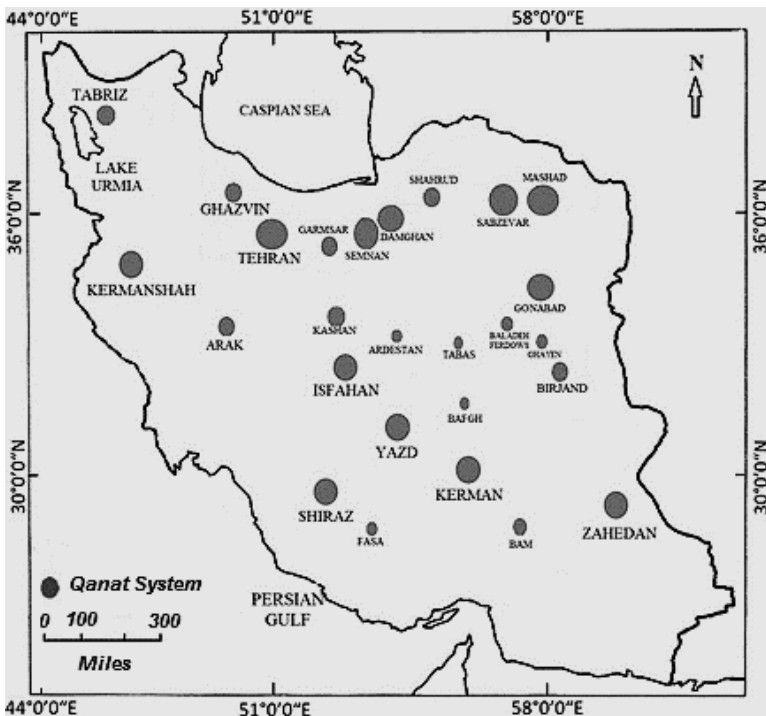

**Figure 5.** The geographical distribution of qanats in Iran

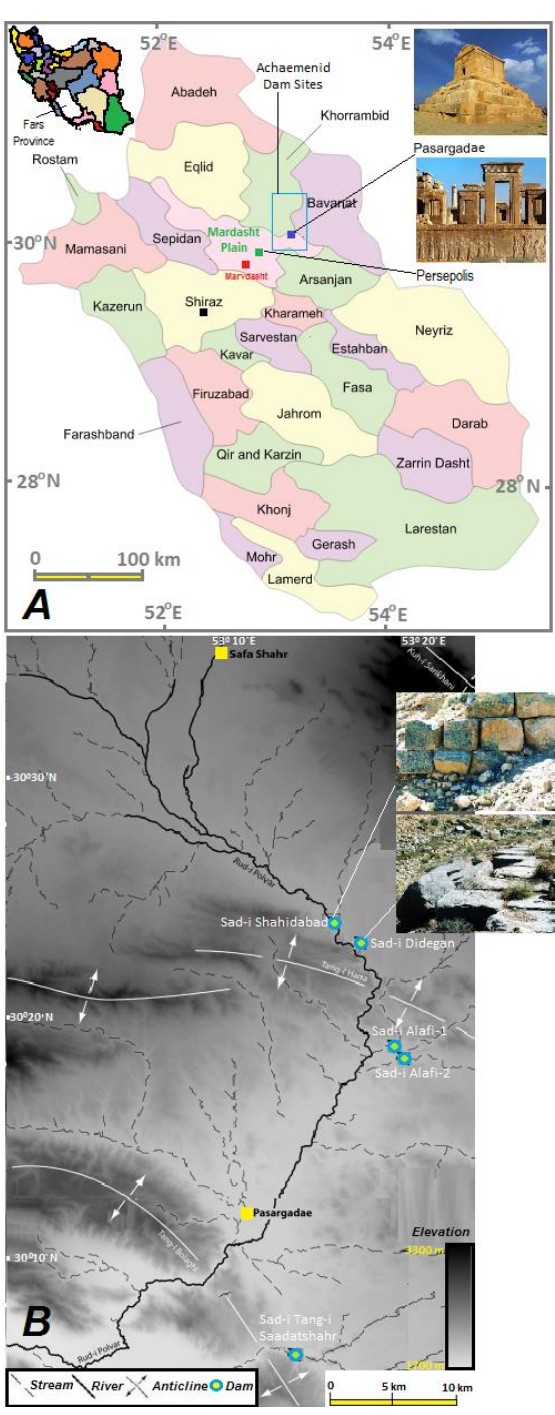

**Figure 6.** Geographical location of Persepolis and Pasargadae (A) and the Achaemenid dams (B) in the Marvdasht Plain, Fars Province (Map B is based on the global SRTM DEM created by Schacht et al., 2012)



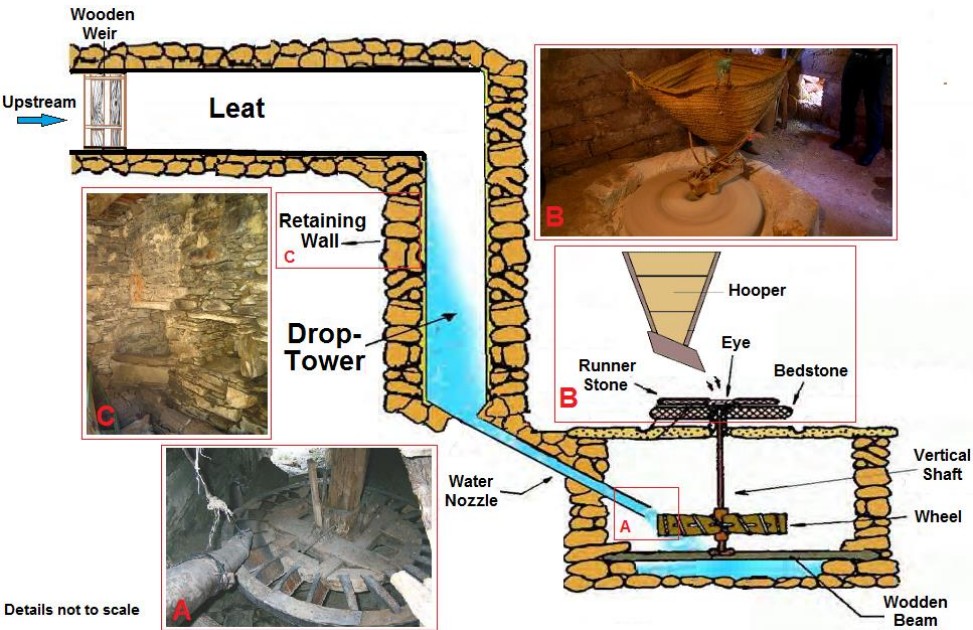

**Figure 7.** Structure of a typical horizontal watermill in Iran

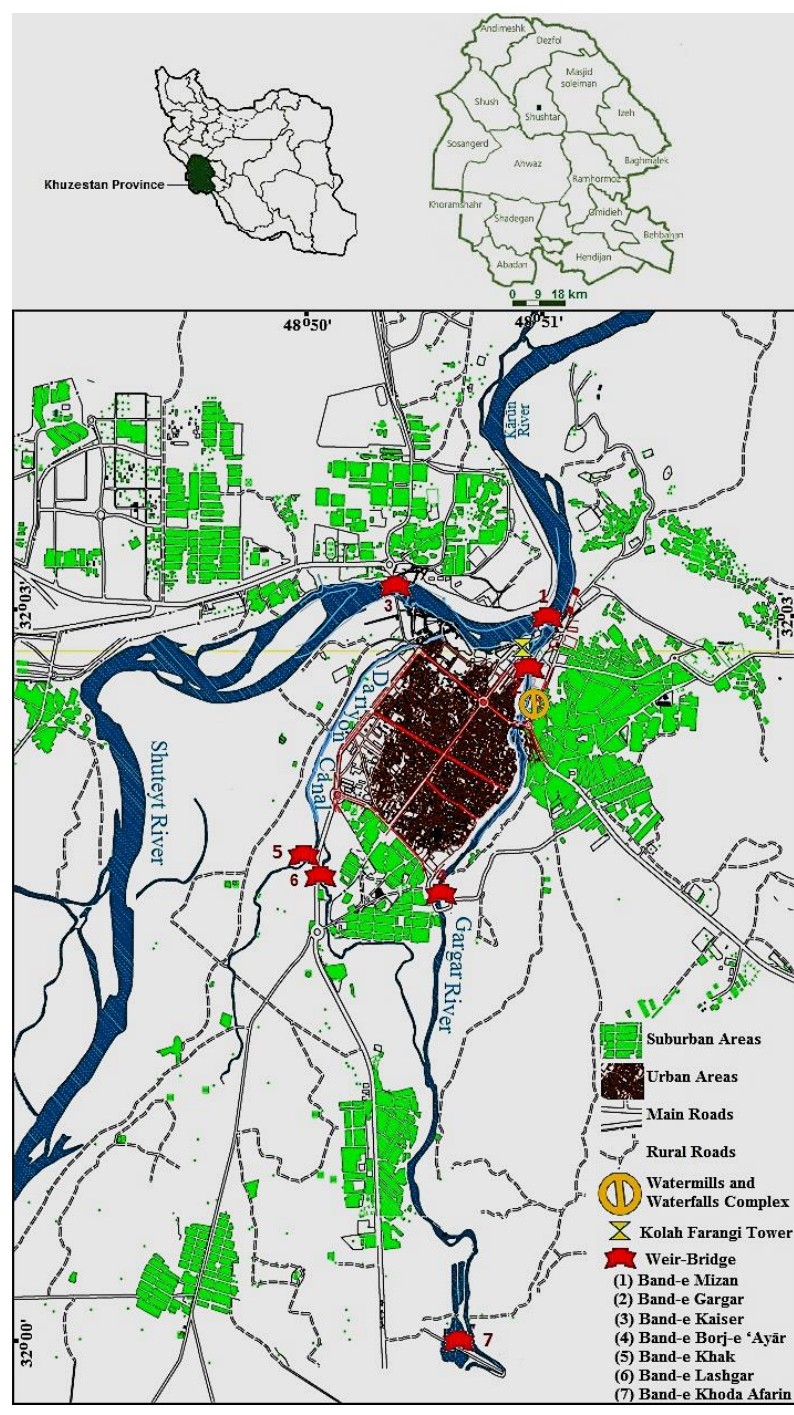

**Figure 8.** Historical hydraulic structures of the Karun River in Shushtar districts, Khuzestan Province (Adopted from UNESCO MAP of Shushtar under CC-BY-SA license)



## Notes

[1] "Irān", meaning "land of the Aryans", has had many changes in its areas. The modern state of Iran is a prominent part of the Iranian Plateau territory and its bordering regions where Iranian culture has had considerable influence. This region is not an existing country and includes much of the Caucasus, Iraq, Afghanistan, Pakistan, and Central Asia. In literature, other names of Iran are "Iranweij", "Eran-Shahr", "Pars region", "Greater Iran", "Persia", and "Iran-Zamin".

[2] Iran is the 18th-largest country in the world by area.

[3] Meters above sea level.

[4] This amount is equal to nearly one-fourth of the world's average rainfall, i.e., ~860 mm/yr.

[5] Kermanshah is also known as "Kermashan", is the Kermanshah Province Capital

[6] Khorramabad is the capital of Lorestan Province.

[7] The Iranians had kept the goat and sheep form ~10,000 years ago. The cattle were domesticated ~8,000 years ago, much earlier than the horse. For more information, see Zeder, M. A. (2001). A metrical analysis of a collection of modern goats (Capra hircus aegargus and C. h. Hircus) from Iran and Iraq: implications for the study of caprine domestication. Journal of Archaeological Science, 28(1), 61-79.

[8] Karun is the largest river by discharge and length in Iran. The length and average discharge of this river are about 950 km and 575 $m^3$/s, respectively.

[9] Karkheh, with a length of about 755 km, is the third major river in Iran that rises from the Zagros Mountains of western Iran, and passes west of Shush (ancient Susa) in Khuzestan Province. In ancient times, this river flowed into the Tigris River close to the Iran-Iraq border.

[10] Jarrahi River is one of the major rivers in southwestern Iran that flows in the provinces of Khuzestan and Kohgiluyeh and Boyer-Ahmad. With a length of 438 km, this river is known as the 11th longest river in Iran.

[11] Dez River, with a length of about 400 km, is the principal tributary of the Karun River.

[12] Susa was the center of the Elam Civilization (2700-539 BC). The ruins of ancient Susa are located in "Shush" modern city in Khuzestan Province in southwestern Iran, near the Iran-Iraq border (east of the Tigris River).

[13] A sealed tank for the collection and temporary storage of sewage.

[14] According to Tamburrino (2010), widespread overexploitation of land and water resources for cultivation in the alluvial plains of Lower Mesopotamia had resulted in siltation and soil salinization.

[15] Hammurabi was the sixth and best-known Babylonian Empire, reigning from 1,792 BC to 1,750 BC.

[16] In the past few centuries, a rural council included "village headman" (in Persian Kadkhoda), "chief water distributor" (in Persian Mirab", and elders (known as or Rish-sefid").

[17] In present-day Fars Province in the south of Iran and north of the Persian Gulf.

[18] Also known as "Persia".

[19] The "semi-nomadic" term is used here to refer to communities that show significant features of both sedentary and nomadic lifestyles, and occasionally to refer to sedentary societies that still maintain a strong historical identity with a past nomadic lifestyle.





---

For more information, see Salsman, T. R. (2019). Constructing Safavid Iran: Space, Pastoralism, Power, and Identity in Safavid Iran 930-1077/1524-1666. The Undergraduate Historical Journal at UC Merced, 6(1).

[20] Many Arian tribes were aware of their affinities, and they spoke various dialects of a mother tongue known as Arian or Iranian (Shahpour Shahbazi 2011).

[21] The present-day Hamadan.

[22] The first dynastic capital and final resting place of Cyrus the Great (550-530 BC).

[23] The Achaemenids' ceremonial capital during the reign of Darius the Great Reign (522-486 BC).

[24] Both sites are referred to as the Marvdasht Cultural Complex.

[25] The "Qanat" term is extracted from Arabic "Qanāh", meaning tube and channel (reef). The word is related to Hebrew "Qāne" and Akkadian "Qanû" with a Semitic (Syro-Arabian languages) root.

[26] In Persian is called "Rahrow" or "Kooreh".

[27] In Persian is called "Mazhar".

[28] In Persian, a skilled qanat digger is called "Muqanni." This profession historically is usually practiced in the family spheres (Kheirabadi 2000). At the beginning of the Achaemenid Empire, the practice of coin was not yet familiar. So, the wages of qanat diggers, rewards, and praise were paid by commodities such as wheat flour, barley, dairy, sheep, fish, eggs, and honey or additional support such as free access to water and land. The first known Persian coins, "Daric" and "Siglos", were introduced by Darius the Great in 514 BC. According to Ghrishman (1987), a tablet' inscription at the Persepolis complex shows only one-third of the wages at the beginning of the Xerxes I's rule (486-465 BC) were paid in cash and toward the end of this period, two-thirds.

[29] Required tools including a well-wheel, container (basket), rope, flashlight, bubble level (spirit level), plummet, string, shovel, and pickaxe.

[30] These records show Sargon II discerned that the defeated city had vibrant and varied vegetation while there was no river to cross. Therefore, he tried to realize the reason why the region could stay green and lush. The answer to this question lay in the existence of qanats. The qanat construction was on "Ursa" orders, the king of the area, who had rescued the people from thirst and turned Uhlu into a prosperous and green land.

[31] A name borrowed from Assyrian sources used for a group of people who lived in a region located between Lake Van and Lake Urumiyeh, named "Urartu".

[32] The king of the Neo-Assyrian Empire (722-705 BC)

[33] In Persian called "Mogh."

[34] The truth behind this is that farmers had given a part of their production and income to priests in the form of firstfruits offering, donation, and gifts, mostly at harvest time.

[35] Pârsa or Pârseh in ancient Persian, Takht-e Jamshid in modern Persian.

[36] Both Pasargadae and Persepolis were regarded as the "birthplace" and "cradle of the Persians."

[37] The present-day Hamadan.

[38] It is comprised of a smaller plain named "Pasargadae Plain."

[39] It is called "Nowrouz," which means "a new day" and is celebrated on the March equinox marking the first spring day, usually occurring on March 21.





---

[40] Also known as "Polvar," "Parvab," and "Sivand."

[41] In Persian "Pole-Khan."

[42] Also known as "Doroodzan Dam."

[43] Some parts of the Didegan Dam body remain sufficiently to guess its original dimensions and materials.

[44] This fastening tool is called "Dom Chelcheleh" (Swallow-tail).

[45] Farvardinegan (the Remembrance Day) is a ceremony to remember and respect the deceased's souls.

[46] Percent of Normal Precipitation Index

[47] Satraps were the governors of the provinces of the Achaemenid Empire.

[48] It was under the control of the Seleucids, but Parthia's Seleucid governor proclaimed his independence. More information is available at Brosius, M. (2006). The Persians. Routledge (Taylor and Francis). Abingdon, U.K.

[49] The Hellenistic period was a time frame from Alexander the Great's death in 323 BC to the emergence of the Roman Empire in 31 BC. For more information, about this period see Hemingway, C., & Hemingway, S. (2007). Art of the Hellenistic Age and the Hellenistic Tradition. The Metropolitan Museum of Art. New York, USA.

[50] "Hecatompylos", also known as "Qumis", was the capital of the Parthians (in present-day Semnan Province)

[51] In Persian, it means "the Bureau of Water Consumption and Production."

[52] In Persian, the" weir" term is called "Band." The main difference between the Sassanids weir and today's dam is that a weir allows water to pass, but a dam does not. Hence, a weir can be used for increasing water level, not water storage purposes.

[53] Shushtar is located in Khuzestan Province. "Sostrate" and "Tustar" are the ancient names of this city.

[54] Dezful is located in Khuzestan Province.

[55] In this time, Khuzestan plains, due to their large wheat, barley, oilseeds fields, and citrus fruits growing, were considered the breadbasket of the Sasanid's Empire. The 1700 years old weirs and water-mills in Dezful and Shushtar were part of the Dez and Karun hydraulic systems. For more information, see Wilkinson, T. J., Boucharlat, R., Ertsen, M. W., Gillmore, G., Kennet, D., Magee, P., Rezakhani, K., & De Schacht, T. (2012). From human niche construction to imperial power: long-term trends in ancient Iranian water systems. Water History, 4(2), 155-176.

[56] "Caesar's Weir", "Valerian Weir," and "Shadorwan Weir-bridge" are other names of "Band-e Kaisar."

[57] At the north of Shushtar, the Karun River is divided into the eastern Gargar River and the western Shuteyt branches. These branches join together in the Band-e Gheer Weir again. The hand-dug Dariyon Channel, with a length of 2.5 km, was excavated the downstream of the Mizan Weir to irrigate the land between the Gargar and Shuteyt rivers." The Dariyon River is also divided into two branches in the Band-e-Khak. The main branch goes towards the south; it joins the Shuteyt River after 33 km in the Arab Hassan Weir. Another branch flows toward the Gargar River.

[58] Gargar, with a length of 80 km to 100 km and a width of 20 m to 90 m, is the most significant human-made watercourse in Iran, which its original construction dates back to the early Sassanian period (Woodbridge et al., 2016). The other names of this river are "Do-Dangeh" and "Mashreghan." The main function of the Gargar was to irrigate agricultural fields in the south of Shushtar and supply water for residential areas. For more information, see UNESCO 2008.

[59] "Shoteit" is derived from "Shatt," an Arabic word meaning "Big River". The other name of this river is "Chahar-Dangeh".

[60] The width of slices is ranging from 1.7 m to 2.85 m.





[61] During 1632 and 1669, Band-e Kaisar and Band-e Gargar were restored by the Safavid governor of Shushtar and Dezful named "Vakhushti Khan Gorji". His son, named "Fathali Khan," who ruled these areas from 1669 until 1694, repaired Band-e Kaisar, but apparently, he made a great mistake. He decided to decrease the Shoteit River discharge by making holes and cracks in the Mizân weir's gates. He thought that the workers could get rid of water and repair the Kaisar Weir effortlessly. This action increased the discharge of the Gargar River. After decreasing the Shuteyt discharge, the farmlands on both sides of this river gradually became dry and unproductive, bringing many negative social and economic consequences for Shushtar and Dezful (Roshani Nia et al., 2007).

[62] In the 18th century, the Mizan and Kaisar weirs, especially their mills, were damaged by a flash flood, causing heavy losses to the economic and social conditions of Shushtar. To solve related problems, these weirs were repaired for three years from 1806 to 1809 by "Mohammadali Mirza Dolat Shah", an Iranian prince of the Qajar Dynasty. For more information, see UNESCO 2008.

[63] In Persian, it is called "Asiyab."

[64] On a small scale (e.g., household level), this process was done by human muscle power.

[65] Windmills extensively appeared in eastern Iran with a dry climate during a time-period between 500 and 900 AD. For more information, see Sharma, R. (2009). Future Power, Future Energy: Wind Power. The Energy and Resources Institute (TERI). New Delhi, India.

[66] Wheat, barley, oilseeds, corn, and occasionally turmeric, and sugar-cane.

[67] "Shapur I" was the second Sasanian King of Kings of Iran who ruled from 240 to 270 AD.

[68] "Shapur II" was the tenth Sasanian King from 309 to 379 AD.

[69] "Kavad I" was the Sasanian King of Kings of Iran from 488 to 531 AD.

[70] "Khosrow I", also known as "Anushirvan", was the Sasanian King of Kings of Iran from 531 to 579 AD.

[71] The water tower diameter differs from a thin 66 cm at "Estahban" water-mill in Fars Province to a wide of 3 m at Shushtar (Harverson 1993).

[72] The nozzle's greater diameter causes a more significant discharge toward the wheel and less time to feed the water tower.

[73] Also known as "Runner Stone."

[74] Known as "Eye."

[75] Millstones in qanats, in comparison with river mills, were remarkably small in diameter.

[76] In literature, there are few disagreements about the horizontal water-mill's efficiency. For instance, Wikander (2000) judged the Greek water-mill technology to be approximately as efficient as the Roman ones, whereas Forbes (1964) stated that the Greek water-mills are less efficient. For more information, see Wikander, Ö. (2000). The Handbook of ancient water technology, 371-400; Forbes, R. J. (1964). Studies in ancient technology. 9 (1964) (Vol. 1). Brill Archive.

[77] The site is Iran's 10th cultural heritage site, registered on the UNESCO World Heritage list (2009).

[78] The water-mill complex is called "Sika."

[79] Dehloran Plain is located in Ilam Province, southwestern Iran, near the Iran-Iraq border.

[80] Jiroft is a city in Kerman Province, south-central Iran.

[81] Nishapur or Nishabur is a city in Khorasan Razavi Province, in northeastern Iran

[82] Located in Esfahan Province.

[83] Located in Esfahan Province.



[84] Located in Yazd Province.

[85] Located in Yazd Province.

[86] Located in Yazd Province.

[87] Kerman is the capital city of Kerman Province.

[88] Located in Fars Province.

[87] These mills were important for qanat owners because some of the rental income from water-mill leases have been spent on qanat's care.

[90] "Zayandeh-Rud" is the largest river of the Iranian Plateau in central Iran.

[91] In this mill, the wheel's bottom is submerged into flowing water, where there are not head differences.

[92] In some parts of Dezful, these are locally known as "Louvineh."

[93] The "Sassanid Bridge" is located in the old part of the city known as the "Qaleh" (castle) Neighborhood.

[94] According to Aarab (2016), Muslim forces first attacked southern Iraq and the plains of Khuzestan, the Sassanid state's political and economic center.

[95] In Arabic, called "Urf."

[96] In Islam, the principles and rules set based on clear and definite texts of the Quran are scant; sharia instead supports common virtuous regulations. It contains several directions expressed in the Quran, augmented through the Sunna (an extensive collection of the Prophet Muhammad's ideas, thoughts, beliefs, morals, manners, and learning-teaching; validated by sayings of the Prophet (called "Hadith"), a unanimous agreement among scholars and religious figures regarding a religious ruling (called "Ijma"), and logical reasoning by analogy (called Qiyas). 'Suitable' local customs (Urf) are identified as background resources of law. Most Islamic communities no longer consider "Ijtihad" (independent reasoning) as a valid mode of legal inquiry, while the Shiite tradition has always accepted "Ijtihad" as a source of law. More information can be found at Al-Awa, M. (1973). The place of custom in Islamic legal theory. Islamic Quarterly, 17, 177–179.

[97] Also known as "Sharia," "Shariah," or "Shari'a."

[98] The Quran is also romanized as the "Qur'an" or "Koran."

[99] Although Arab Muslims allowed farmers to own their land, qanat, and well, they divided the Iranians into Muslims and non-Muslims. Muslims had to pay taxes, but non-Muslims had to pay Jizyah in addition to taxes. For this reason, many non-Muslim Iranians were forced to leave their lands and migrate to neighboring regions such as India.

[100] One difficulty was the decline of irrigation agriculture throughout the "Dark Ages of the Sasanians", which resulted in flash floods, which washed away croplands, damaged water infrastructure, and threatened food security, safety, and the economy of the territory.

[101] This period in Iran starts with the rise of the Samanids and ends with the fall of the Khwarezmians and Mongols' arrival (1098 to 1219 AD).

[102] During the Abbasid caliph Harun al-Rashid (786 to 809 AD), the Islamic government strongly patronized scholars. After the foundation of the House of Wisdom (in Arabic Bayt Al Hikma) in Baghdad, scholars from different parts of the world were tasked to collect and translate all of the classical knowledge of the day into Arabic and then to Persian and Turkish. Although Islam's Golden Age begun in Baghdad and developed in Islamic regions, it was not just the outcome of Islamic achievements.





---

[103] Today, this book is available in French, Italian, and English languages.

[104] Also known as Abu Ali Sina.

[105] This book was written in Arabic, the official language, and then translated into different languages.

[106] Also known as "Chronology of Ancient Nations" or "Vestiges of the Past".

[107] According to Sorkhabi (2017), Biruni, in his book entitled "Alberuni's India." documented a hypothesis about the artesian phenomenon as follows: "*The elevation of the Waterhouse (aquifer) containing hidden water (groundwater) is higher than the elevation of the artesian well to allow water's flowing out. If the elevation of the Waterhouse is high enough, the water could easily flow to the top of the surface*" (Biruni ~1030 AD). These ideas confirm Biruni and other scholars at their level were aware of the concepts applied in advanced geology, hydrology, and hydrogeology. For more information, see Sorkhabi, R. (Ed.). (2017). Tectonic Evolution, Collision, and Seismicity of Southwest Asia: In Honor of Manuel Berberian's Forty-five Years of Research Contributions (Vol. 525). Geological Society of America, USA.

[108] In present-day Shiraz, Fars Province.

[109] In present-day Tous in Razavi Khorasan Province.

[110] 23 km southeast of present-day Qom, the capital of Qom Province.

**Appendix**




**Table A1** Average and moving average of precipitation for three, five and seven years from 1973 to 2016

| Year | Annual Rainfall (P) (mm) | P-$\bar{P}$ (mm) | 3 years moving average of rainfall (P3) (mm) | P3-$\bar{P}$ (mm) | 3 years moving average of rainfall (P3) | P5-$\bar{P}$ | 3 years moving average of rainfall (P3) | P7-$\bar{P}$ |
|---|---|---|---|---|---|---|---|---|
| 1973 | 131 | -212 | | | | | | |
| 1974 | 281 | -62 | 260 | -84 | | | | |
| 1975 | 368 | 24 | 371 | 27 | 286 | -58 | | |
| 1976 | 464 | 120 | 339 | -5 | 329 | -15 | 324 | -20 |
| 1977 | 184 | -159 | 332 | -12 | 371 | 27 | 336 | -8 |
| 1978 | 348 | 4 | 341 | -3 | 340 | -4 | 335 | -9 |
| 1979 | 489 | 146 | 351 | 7 | 302 | -42 | 334 | -10 |
| 1980 | 214 | -129 | 326 | -18 | 338 | -6 | 334 | -10 |
| 1981 | 273 | -70 | 283 | -61 | 362 | 18 | 346 | 2 |
| 1982 | 361 | 17 | 368 | 24 | 316 | -28 | 367 | 24 |
| 1983 | 468 | 124 | 364 | 20 | 374 | 30 | 353 | 9 |
| 1984 | 262 | -81 | 411 | 67 | 396 | 52 | 382 | 38 |
| 1985 | 502 | 158 | 384 | 40 | 408 | 64 | 392 | 49 |
| 1986 | 386 | 42 | 436 | 92 | 383 | 39 | 384 | 40 |
| 1987 | 419 | 75 | 384 | 40 | 391 | 47 | 359 | 15 |
| 1988 | 346 | 2 | 356 | 12 | 350 | 6 | 353 | 9 |
| 1989 | 301 | -42 | 315 | -29 | 317 | -27 | 324 | -20 |
| 1990 | 295 | -48 | 273 | -71 | 293 | -51 | 315 | -29 |
| 1991 | 221 | -122 | 272 | -72 | 288 | -56 | 314 | -30 |
| 1992 | 299 | -44 | 280 | -64 | 309 | -35 | 327 | -17 |
| 1993 | 320 | -23 | 343 | -1 | 338 | -6 | 313 | -30 |
| 1994 | 410 | 66 | 390 | 46 | 335 | -8 | 324 | -20 |
| 1995 | 439 | 96 | 353 | 9 | 350 | 6 | 345 | 2 |
| 1996 | 207 | -136 | 340 | -4 | 360 | 16 | 354 | 10 |
| 1997 | 371 | 27 | 316 | -28 | 350 | 6 | 386 | 43 |
| 1998 | 369 | 25 | 367 | 23 | 371 | 27 | 356 | 12 |
| 1999 | 359 | 15 | 425 | 82 | 368 | 25 | 372 | 28 |
| 2000 | 547 | 203 | 367 | 23 | 405 | 61 | 421 | 77 |
| 2001 | 194 | -149 | 431 | 87 | 442 | 98 | 405 | 61 |
| 2002 | 552 | 208 | 434 | 90 | 421 | 77 | 419 | 76 |
| 2003 | 554 | 210 | 454 | 110 | 406 | 62 | 418 | 74 |
| 2004 | 254 | -89 | 427 | 83 | 437 | 93 | 367 | 23 |
| 2005 | 473 | 129 | 359 | 15 | 364 | 20 | 371 | 27 |
| Year | Annual Rainfall (P) (mm) | P-$\bar{P}$ (mm) | 3 years moving average | P3-$\bar{P}$ (mm) | 3 years moving average | P5-$\bar{P}$ | 3 years moving average | P7-$\bar{P}$ |


| | | | of rainfall ($P_3$) (mm) | | of rainfall ($P_5$) | | of rainfall ($P_7$) | |
|---|---|---|---|---|---|---|---|---|
| 2006 | 348 | 4 | 337 | -7 | 298 | -46 | 357 | 13 |
| 2007 | 188 | -155 | 254 | -90 | 338 | -6 | 326 | -18 |
| 2008 | 225 | -118 | 290 | -54 | 310 | -34 | 348 | 4 |
| 2009 | 455 | 111 | 338 | -6 | 322 | -21 | 353 | 9 |
| 2010 | 333 | -10 | 400 | 56 | 387 | 43 | 351 | 7 |
| 2011 | 409 | 65 | 418 | 74 | 409 | 65 | 374 | 30 |
| 2012 | 511 | 167 | 418 | 74 | 387 | 43 | 356 | 12 |
| 2013 | 332 | -11 | 397 | 53 | 341 | -3 | 320 | -24 |
| 2014 | 348 | 4.1 | 262 | -82 | 299 | -44 | | |
| 2015 | 103 | -240 | 218 | -126 | | | | |
| 2016 | 201 | -142 | | | | | | |

($\overline{P}$) is the mean annual rainfall from 1973 to 2016 (343 mm/y)