# Peer review of "Water Resources Management, Technology, and Culture in Ancient Iran"

_Hydrology and Earth System Sciences, 2021_

## Author Comment (AC1)

Masoud Saatsaz
Institute for Advanced Studies in Basic Sciences
(IASBS), Zanjan, Iran
Phone: +98 33153777
Mobile: +98 916 615 7603
e-mail: saatsaz@iasbs.ac.ir

Date: 30.07.2021

Dear Reviewer,

We appreciated the time that you have taken in our manuscript and the constructive comments.

In this revised version of the manuscript, we have tried to do our best to address your comments. We have included many paragraphs in many parts of the manuscript to improve our contribution. A slight change has been made in the title to be more engaged with the body. We also tried to change titles and sub-titles to reflect our intentions and visions clearly. More specifically, we have rewritten some sections and added more information to enhace the quality.

In what follows, the comments and our replies are in simple text, respectively.

We hope this version will be considered positively for publication.

Thank you again for your consideration.

Sincerely,

Masoud Saatsaz

Corresponding Author

*Reviewer #1: comments*
*I appreciate the authors for their efforts to shed light on such an important subject, but their work has plenty of room for improvement.*

   1- *The article title is both ambitious and ambiguous. "Water Resources Management, Technology and Culture in Ancient Iran" makes the readers expect to learn about an argument on a kind of relationship between technology and culture which has emanated from water resources management. However, throughout the article, the readers do not come across such a reasoning in an attempt to prove that water resources management served as a historical context in which technology and culture intersected or interacted.*

Answer: First of all, we would like to thank you for your high-quality reviews of our manuscript and your careful comments.

We agree with you. We think it was not necessary to cite the word "sources" in the title. We have tried to sort the terms "Technology, Management, and Culture" according to their engagement in the manuscript.

In this version, we have changed the article's title as follows:

Old title: Water Resources Management, Technology and Culture in Ancient Iran

New title: Technology, Management, and Culture of Water in Ancient Iran

2- The topic *"Historical Evolution of Life on the Iranian Plateau"* does not correspond to the historical period of this study, but it's more of a short introduction that goes back to as early as 90000 years ago and then ends with the advent of agricultural communities in Iran. This topic could have been an introduction to the general history of Iran over the same period of time as that of this study, in view of water issues. This way, the readers could have gained a general knowledge that is precursor to the next discussion. Talking about the migration of Homo Sapience into the Iranian plateau and the history of plant domestication shows little relevance to the main argument. Moreover, Rihel and his colleagues claim that the first farming communities appeared on the Iranian plateau even earlier that 7 or 8 millennia BC as follows: Rihel, S., Zeidi, M., & Conrad, N. J. (2013). Emergence of agriculture in the foothills of the Zagros mountains of Iran. Science, Vol. 341, 65-66.

Answer: This section has been totally revised as follows:

Old version:
3. Historical Evolution of Life on the Iranian Plateau

Living on the Iranian Plateau started with the dispersal of early modern humans from Africa, dated between at least ~90,000 and ~50,000 years ago in the Middle-Paleolithic of the Stone Age (Delson 2019). The oldest-known artifacts from the Middle-Paleolithic, such as stone tools have been found at the Varvasi Cave (in Kermanshah ), Yafteh Cave (around Khorramabad ), Kashafrud (in Khorasan Razavi), and Ganj Par Site (in Rostam Abad, Gilan), signifying the human existence on the Iranian Plateau (Vigne et al., 2005). During the 8th and the 7th Millennium BC, the first agricultural communities started to emerge in southwestern, western, and northwestern Iran, where perennial rivers, rainfall, and fertile alluvial soils allowed agrarian societies to develop (Riehl et al., 2013). Meanwhile, the earliest animal domestication began to occur in the Taurus and Zagros Mountains  (Zeder 2008; Helmer et al., 2005). Agriculture and domestication growth caused some community members to engage in off-farm activities such as construction, mining, woodworking, metalworking, trading, stone cutting, and other services. All of these components were essential for early civilizations to emerge (Mountjoy 2005).

New version:
3. The Water Significance in Prehistoric Iran

Living on the Iranian Plateau started with the dispersal of early modern humans from Africa, dated between at least ~90,000 and ~50,000 years ago in the Middle-Paleolithic of the Stone Age (Delson 2019). The oldest-known artifacts from the Middle-Paleolithic, such as stone tools, have been found at (i) the Varvasi Cave in the Dinurab River Basin , (ii) Yafteh Cave in the Khorramabad River Vally , (iii) Kashafrud  Site along the Kashafrud River Basin , and (iv) Ganj Par Site around the Sefidrud River , signifying the human existence in water-rich regions of western and north of Iran (Vigne et al., 2005). During the 8th and the 7th Millennium BC, the earliest villages (e.g., Chogha Bonut , Ali Kosh , Ganj Dareh , and Teppe Sarab ) early agricultural communities started to emerge in southwestern, western, and northwestern Iran, where perennial rivers, rainfall, and fertile alluvial soils allowed agrarian societies to develop (Alizadeh 2003; (Zeder and Hesse 2000; Riehl et al., 2013; Potts 2014; Gallego-Llorente et al., 2016). The sedentary lifestyle introduction resulted from wild plant gathering by the early Neolithic hunters and gathers. Meanwhile, the earliest animal domestication began to occur in the Taurus and Zagros Mountains  (Zeder 2008; Helmer et al., 2005). In the early stage of domestication, nomadic pastoralism was practiced in southwestern Iran (Zeder and Hesse 2000; Misra 2009). Agriculture and domestication growth caused some community members to engage in off-farm activities such as construction, mining, woodworking, metalworking, trading, stone cutting, and other services. Indeed, it was during a period between 6500 and 3800 BC that core regions formed in Iran in the light of artificial water management, characterized by developing water canal networks and breaching levee-banks (Adams, 1965; Hole et al., 1987; Gillmore et al., 2009).

As you can see, many references has been added in the context from reliable sources to back up claims.

3- In the article, there are some contradictions and conceptual conflicts between some facts mentioned in the different parts of the article. Also, some controversial hypotheses are treated as definite proven facts. For example, on page 10, in the first paragraph, the authors contend that Islamization of water rules was impeded and stopped by some barriers, whereas immediately in the next paragraph they talk about the Islamic Sharia and its rulings about water issues. Also, on the same page, the authors take it for granted

*that qanat has been native to Iran and spread from Iran to its neighboring countries, though this subject is still controversial and it's very difficult to pinpoint any given area as the actual cradle of qanat system.*

Answer: in this version we tried to remove the available contradictions in the manuscript. For example, about the qanat origin, we have changed this sectence as follows:

Old version:

Immediately after the arrival of Islam, Iran had a messy and disorganized environment. Muslims tried to change the religious, political, institutional, and social structure of the country. *The implementation of Islamic customs and laws was one of the first steps towards the Islamization of the society. In the meantime, water could be an essential link between custom, religion, law, and community, but there were obstacles problems in the Muslims' path.* In the sources of sharia, there were only some concepts such as justice, fairness, and balance, for the benefit of all societies (Naff 2009). Although the Quran has 63 references to water (Farshad and Zinck 1998), it does not assert any clear duty or rule on water supply and consumption (Absar 2013). The lack or insufficiency of fundamental rights and obligations regarding access to water, sanitation, sharing, and selling water was the main barrier to the Islamization of water-related rules. As a case, Arab Muslims had no law or regulation about qanats because qanat was native to Iran and spread from Iran to neighboring countries.

New version:

Immediately after the arrival of Islam, Iran had a messy and disorganized environment. Muslims tried to change the religious, political, institutional, and social structure of the country. *In general cases, the implementation of Islamic customs and laws had been considered as one of the first steps towards the Islamization of the society. However, there were obstacles in the Muslims' path to make changes in water-related fields. In the initial sources of sharia (e.g., Quran),* there were only some concepts such as justice, fairness, and balance, for the benefit of all societies (Naff 2009). Although the Quran has 63 references to water (Farshad and Zinck 1998), it does not assert any clear duty or rule on water supply and consumption (Absar 2013). As a case, Arab Muslims had no law or regulation about qanats because this system and its culture *was developed* in Iran and introduced subsequently to neighboring countries.

As you can see, most of these contradictions were due to improper selections of words that have changed the meaning of these sentences to something completely different from their intents.

4- *The article needs more integration. The facts provided in the article are not very interrelated and interconnected.*

Answer: We have attempted to add sentences or paragraphs, improving logical flow in the manuscript body. Also, we have organized and changed all heading or subheading titles to separate distinct parts of the manuscript.

5- *The article cannot come up with a novel fact, discovery or interpretation. The authors have done a great job to read through many sources and glean a valuable set of information, but all they say is in fact a different rehash of our previous knowledge. The article is not expected to produce new facts or present new discoveries from scratch, but at least it could abstract a novel historical pattern or suggest a new interpretation about a historical link between hydraulic technologies and Iranian culture, by juxtaposing those diverse historical facts.*

We thank you for your appreciation of our work. About the research novelty, we tried to enter our interpretation and attitude in the body. Some parts of the article are totally novel. Some discussions about the relationship between water technology and culture (e.g., the role of water technology in holding Persian Norouz celebration) are not enough discussed in other researches. Our calculation about the impact of drought on the Persian communities that justifies the Achaemenid concerns about the drought or showing the role of water celebrations to encourage people to value, respect, and protect water (added in the new version) are completely original. We have attempted to add a conclusion section at the end of each section to explain our findings. Sometimes, we tried to provide a new vision for our readers by merging or adding more details. Also, it is too hard to find a similar case to compare. In the new version, we also have enhanced the conclusion by addressing our findings.

---

## Author Comment (AC2)

Masoud Saatsaz
Institute for Advanced Studies in Basic Sciences
(IASBS), Zanjan, Iran
Phone: +98 33153777
Mobile: +98 916 615 7603
e-mail: saatsaz@iasbs.ac.ir

Date: 31.07.2021

Dear Reviewer,

We appreciated the time that you have taken in our manuscript and the constructive comments.

In this revised version of the manuscript, we have tried to do our best to address your comments. We have included many paragraphs in many parts of the manuscript to improve our contribution. A slight change has been made in the title to be more engaged with the body. We also tried to change titles and sub-titles to reflect our intentions and visions. More specifically, we have rewritten some sections and added more information to enhance the quality.

We hope this version will be considered positively for publication.

Thank you again for your consideration.

Sincerely,

Masoud Saatsaz

Corresponding Author

*Reviewer #1: comments*
*I appreciate the authors for their efforts to shed light on such an important subject, but their work has plenty of room for improvement.*

1- *The paper's subject is unclear. The title suggests that the authors will argue the specific way(s) in which technology and culture interacted in shaping water management in ancient Iran. Nevertheless, this topic is not touched on in the paper. The Abstract suggests two other topics: a) a brief history of water management in Iran; b) how geo-climatic functions control water regimes, settlement patterns, and [undefined] socio-economic issues. A brief history of water management in Iran fits most closely the content of the paper.*

Answer:

First of all, we would like to thank you so much for your high quality of editing.

You are right. This work can be considered as a historical investigation. As we mentioned before to Prof. Ertsen, Dr Saatsaz has published an article titled "A historical investigation on water resources management in Iran" in 2019. This article is quite different from that of 2019. For this reason, we tried to select a different title for this manuscript.

That research was related to water resources management throughout different traditional, transitional, and modern periods. In that article, water management, technology, and culture in ancient Iran were not discussed much. As he was writing that article, he realized there are many other essential issues on water resources governance, hydraulic engineering, management practices, and cultural issues in ancient Iran to consider. As far as we know, no article thoroughly explained the whole topics mentioned above. Several scholars have studied the development and management of water resources in ancient Iran. One of the limitations of most previous studies is focusing on either water-related technological systems or socioeconomic aspects of water. One of the main characteristics of

this study is the description of a set of technical, management, and cultural topics, from pre-civilization times to the end of the Islamic Golden Age (1219 AD), and making the link among them, if possible. This task was not easy; it was done by reading more than three hundred articles, books, documents, and maps over three years.. This task was not easy; it was done by reading more than three hundred articles, books, documents, maps over a period of about three years.

In this version, we have changed the article's title and sort its terms "Technology, Management, and Culture" according to their engagement in the manuscript. As follows:

Old title: Water Resources Management, Technology and Culture in Ancient Iran

New title: Technology, Management, and Culture of Water in Ancient Iran

2- *I was not able to find an original argument in the paper. A very general introduction with ambitious statements is followed by detailed facts about individual hydraulic structures across the country. Those facts, if well integrated, could be used in a survey-style textbook chapter on hydraulic structures in ancient Iran. The research questions stated in the introduction (48-52) can be considered an outline of a term paper, but they fall short of proposing or addressing a worthwhile problem in water history studies.*

Answer:

Concerning the items above, some sections have been totally revised in the items above. Some discussions have been added, and some new subjects have been included in the manuscript to improve logical flow and transitions between the sections. Also, we have outlined all heading or subheading titles to separate distinct parts of the manuscript.

Regarding the proposal for publishing this work as a chapter book, we believe this work in the format of an article in this valuable journal can be more accessible for Iranian researchers. Most Iranian scholars cannot buy an original international book due to their economic situation and lack of access to international banks. However, this explanation is not a scientific reason and does not justify reducing our efforts to improve the research.

3- *The facts about specific hydraulic infrastructures and specific archaeological sites are too detailed without being integrated into a general theoretical discussion. Literature review without integration has resulted in some contradicting statements, for example, regarding the state of water management in the Islamic period. A statement about the innovations in hydraulic technology in the early Islamic period is followed by a paragraph on the demise of water management in the same period. For the same reason, some sections like "Historical Evolution of Life in Iran" do not relate to the paper's topic.*

Answer:

As mention in the previous answer, we tried to tackle these concerns through stated above ways. For example, concerning the item related to the section "Historical Evolution of Life in Iran," we have changed the title and rewritten the entire paragraph to confirm that the first Pre-Iranians lived alongside rivers do in the present.

4- *Several critical arguments in the paper are wrong, outdated, or unsupported. To name a few:*

*There is no evidence that difficulties of access to water management for a large urban population in southwestern Iran were the catalyst of systematic water management. It is unclear what do the authors mean by systematic water management. In any case, early riverine societies seem to have developed canal irrigation independently and simultaneously.*

Answer:

In this manuscript, we have mentioned:

"systematic development appeared in west and southwest Iran closely related to Mesopotamia."

In the above sentence, "systematic" means an action that affected the entire society on a relatively large scale.' Through a systematic development in this area, the first proto-cities of the Middle East, such as "Susa," began to flourish around the Uruk period (4000 to 3100 BC), a prehistoric period of Mesopotamia. Undoubtedly, if there was no systematic development, these cities did not merge or vanished after a time. However, other non-systematic small scall actions were not at a level that their effects can be accurately quantified.

5- *There is no evidence that the irrigation systems of southwestern Iran were the adoption of a Mesopotamian water management system. The entire discussion on "Mesopotamian water management" is inaccurate. Canal irrigation was not an inherently Mesopotamian development. Nor was Fertile Crescent's agriculture and economy under Mesopotamian control.*

Answer:

Many reasons show Mesopotamia influenced the irrigation systems of southwestern Iran. The main reason can be finding one of the earliest water-related regulations issued by King Hammurabi, one last of the Mesopotamian Empires ((who established the Old Babylonian state). Another reason is that the nearest and most vital historical region to the south of Iran was Mesopotamia.

6- *Even though qanat irrigation is a much later technology than canal irrigation, and even though qanat tends to be found in arid regions without permanent rivers, one cannot simply state that qanats were developed because the "Mesopotamian water-based technology" could not meet the needs of arid regions. This statement presumes a causal relationship between two separate phenomena with their complicated development history.*

Answer:

We changed this sentence to be more logical as follows:

Old version:

By the 4th Millennium BC, while water access became more difficult as population growth, economic activity, and urbanization progress, water resources' systematic development appeared in west and southwest Iran under the Mesopotamian civilization. However, despite all benefits, Mesopotamian water-based technology and administration could not meet all water demands in Iran's arid regions.

New version:

By the 4th Millennium BC, while water access became more difficult as population growth, economic activity, and urbanization progress, systematic development appeared in west and southwest Iran closely related to Mesopotamia. However, existing technologies and administration and administration could not meet entire water demands in all Iran's arid regions.

Undoubtedly, by increasing population and the need for more food, qanats were welcomed in Central Iran. After developing qanats, whole season cropping associated with growing various fruits and crops expanded throughout Iran. Other qanat benefits and the role of the Achaemenid administration in water governance have been mentioned in many parts of the manuscript.

7- *The origin and early history of qanat's development are still unknown. It is hard to pinpoint its development to a specific location even though Iran seems to be a good candidate. Moreover, there is absolutely no data to support that Achaemenids used qanats to intensify agriculture across their empire, other than the Western desert in Egypt. In fact, the most well-dated evidence from the Achaemenid period from Fars is for canal irrigation.*

Answer:
In this manuscript we tried to show that the success of the Achaemenidsin water resources development was due to these factors:

1- Proper adimnestration

     2- Water infrastructure such as dams, qanats, canal networks
     3- Public corporation

All these factors were interrelated, but each had its own role. Hence, many paragraphs have been provided to make a bridge between these factors and support the claims about their importance.

    8- *There is no evidence to support recession in irrigation or agriculture in the Seleucid or Parthian period. The most well-documented evidence comes from Khuzistan which shows a steady development of canal systems from Achaemenid to the Sasanian period.*

Answer:

Thanks for your comment. We have made substantial changes in this section as follows:

Old version:

**6. Water Resources Management in the Seleucids Era and Parthian Era**
Following the conquest of Iran by Alexander the Great in 330 BC, the Iranian satraps[i] were governed by various Greek Satraps forming the Hellenistic Seleucid Empire and then the Parthian Empire[ii] (Curtis 2007). After the conquest of Iran by Alexander, qanats seem to have been abandoned or destroyed (Ashrafi and Safdarian 2015). Moreover, since the Parthian government was remarkably decentralized, the Parthians were not concerned about the loss of qanats and other hydraulic structures. According to Semsar Yazdi (2006), some qanat systems and irrigational systems were abandoned or damaged. Polybius, a Greek historian of the Hellenistic period[iii], recorded that Arsaces III, one of the Parthian kings, tried to destroy some qanats and interrupt water flow to make it difficult for the Seleucids to advance towards the Parthian capital[iv] (Beaumont 1971).

**6. Water in the Seleucids Era and Parthian Era**

Following the conquest of Iran by Alexander the Great in 330 BC, the Iranian satraps[v] were governed by various Greek Satraps forming the Hellenistic Seleucid Empire and then the Parthian Empire[vi] (Curtis 2007). In this era, Iran was nominally a united country; it was composed of some semi-independent and sometimes scattered states. The central government did not interfere in the internal affairs of the states except in cases related to security and peace. Besides, the Parthian emperors did not promote a single religion. Hence, there were no single and fixed judicial principles. These factors reduced consensus, unity, and cooperation among Iranians. In the early Parthian period, the Parthians could not manage water-related structures like the Achaemenids could. Qanats and other water-related facilities seem to have been abandoned or destroyed due to internal strife[vii] and wars with Rome (Ashrafi and Safdarian 2015). According to Wenke (1981), agricultural development in the first two centuries AD was concentrated in certain well-watered regions. Subsequently, the agricultural activities decreased in water-scarce regions due to poor water resource management, causing environmental degradation, a decline in rural family income, a rise in rural unemployment, and growth in rural-urban migration. Along with the migration, the number of urban centers increased; urbanization changed society and the economy. At this time, trade and manufacture activities reached their peaks[viii]. Later, the Sassanids made these factors more complete and purposeful by expanding agriculture and creating a suitable administrative organization.

    9- *The conclusion is a very general discussion of modern water management issues in Iran rather than a summary of the body of the evidence.*

Answer:

We have made substantial changes in several parts of the conclusion to address the reviewer's comment.
* * *
[i] Satraps were the governors of the provinces of the Achaemenid Empire.

[ii] It was under the control of the Seleucids, but Parthia's Seleucid governor proclaimed his independence. More information is available

at Brosius, M. (2006). The Persians. Routledge (Taylor and Francis). Abingdon, U.K.

iii The Hellenistic period was a time frame from Alexander the Great's death in 323 BC to the emergence of the Roman Empire in 31 BC. For more information, about this period see Hemingway, C., & Hemingway, S. (2007). Art of the Hellenistic Age and the Hellenistic Tradition. The Metropolitan Museum of Art. New York, USA.

iv "Hecatompylos", also known as "Qumis", was the capital of the Parthians (in present-day Semnan Province)

v Satraps were the governors of the provinces of the Achaemenid Empire.

vi It was under the control of the Seleucids, but Parthia's Seleucid governor proclaimed his independence. More information is available at Brosius, M. (2006). The Persians. Routledge (Taylor and Francis). Abingdon, U.K.

vii Polybius, a Greek historian of the Hellenistic period, recorded that Arsaces III, one of the Parthian kings, tried to destroy some qanats and interrupt water flow to make it difficult for Seleucids to advance towards the Parthian capital (Beaumont 1971).

viii Improved transportation, wel1-developed coinage systems, and opening the Silk Road played significant roles in trade development in this period.